# WHEN TO RETRAIN A MACHINE LEARNING MODEL

## ABSTRACT

A significant challenge in maintaining real-world machine learning models is responding to the continuous and unpredictable evolution of data. Most practitioners are faced with the difficult question: when should I retrain or update my machine learning model? This seemingly straightforward problem is particularly challenging for three reasons: 1) decisions must be made based on very limited information - we usually have access to only a few examples, 2) the nature, extent, and impact of the distribution shift are unknown, and 3) it involves specifying a cost ratio between retraining and poor performance, which can be hard to characterize. Existing works address certain aspects of this problem, but none offer a comprehensive solution. Distribution shift detection falls short as it cannot account for the cost trade-off; the scarcity of the data, paired with its unusual structure, makes it a poor fit for existing offline reinforcement learning methods, and the online learning formulation overlooks key practical considerations. To address this, we present a principled formulation of the retraining problem and propose an uncertainty-based method that makes decisions by continually forecasting the evolution of model performance evaluated with a bounded metric. Our experiments addressing classification tasks show that the method consistently outperforms existing baselines on 7 datasets. We thoroughly assess its robustness to varying cost trade-off values and mis-specified cost trade-offs.

## 1 INTRODUCTION

In many industrial machine learning settings, data are continuously arriving and evolving (Gama et al., 2014). This means that a model, $f_\theta$, that was trained on a fixed dataset, $\mathcal{D}$, will become outdated. This usually translates to a cost in the form of a missed opportunity. However, retraining a new model, $f_{\theta'}$, on a more up-to-date dataset, $\mathcal{D}'$, is also costly. Beyond the obvious costs of computational resources and energy (Strubell et al., 2020), there are human resource costs associated with assigning experts to deploy and maintain the model, as well as collecting and cleaning data. Deploying a new model also generally comes with a higher risk. Therefore, the optimal retraining schedule depends on this comprehensive cost of retraining, on the cost of making mistakes, and on future model performance. Figure 1 provides a visualization of the task.

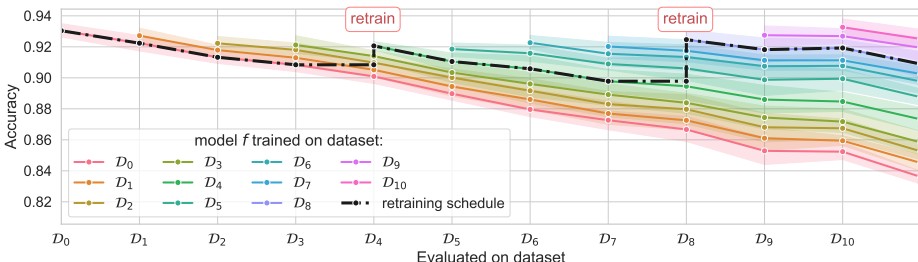

Figure 1: The Retraining Problem: The performance of a model trained on a dataset $\mathcal{D}_i$ gradually decreases when evaluated on more recent datasets in the presence of distribution shift. The task is to determine when retraining is beneficial compared to keeping an older model. We must take into consideration the trade-off between potential accuracy gains and the costs associated with retraining. In the training schedule $\boldsymbol{\theta}$ shown here, retraining occurs twice, at $t = 4$ and $t = 8$.

Although this retraining problem is ubiquitous in industry (Gama et al., 2014), there are few works in the machine learning literature that tackle it directly. It has been framed as an application of the distribution shift detection problem (Bifet & Gavaldà, 2007), where the conventional strategy involves triggering retraining whenever a substantial shift is detected (Bifet & Gavaldà, 2007; Cerqueira et al., 2021; Pesaranghader & Viktor, 2016). However, this approach overlooks retraining costs. This can be particularly problematic when training is expensive, as demonstrated in our experiments. Others have reduced the need for retraining by incorporating robustness to distribution shifts (Schwinn et al., 2022) or adapting to them (Filos et al., 2020), but these methods have limits on the extent of the shift they can handle. Other related areas are online or adaptive learning (Hoi et al., 2021) and life-long learning which updates models with a continuous stream of data through gradual gradient updates, and transfer learning which adapts model from one distribution to another. However, this differs from our problem, as it focuses on maximizing performance while abstracting the practical retraining costs involved in production deployment. In practice, the cost of retraining can go beyond the number of gradient updates or sample complexity, as discussed above. Finally, because this is a sequential decision problem, it can be framed within the offline reinforcement learning framework (Levine et al., 2020). In theory, offline RL methods should be applicable, but few, if any, are designed for very low-data settings. They require substantial amounts of data for training and hyperparameter tuning, and are therefore largely unsuitable to use in this context.

A direct treatment of the cost consideration in the retraining problem is presented by Žliobaitė et al. (2015) and by Mahadevan & Mathioudakis (2024). The formulation by Mahadevan & Mathioudakis (2024) accounts for the trade-off between the cost of retraining and the cost of performance. Their method, CARA, relies on approximating the performance of a model on new data, and the retraining decision is based on this value. However, this approach makes several limiting assumptions: 1) the relative cost objective assumes that the "difficulty" of the task remains constant; and 2) the performance approximation assumes the data distribution is almost stationary.

Instead, we consider a more general objective that combines both the retraining cost and the average performance over a specified horizon. We detail the relationship between our objective and CARA's objective in Appendix 8.10. Our formulation is more general and does not depend on strong assumptions regarding the data distribution and its impact on performance. There is no constraint on how the "retrained" model is obtained. It can be obtained through fine-tuning from a previous model, adapted, trained from scratch, or any other procedure. Additionally, our method can leverage new observations of the model's performance. Our proposed method involves forecasting the performance of both future and current models and making decisions based on the uncertainty of our predictions. We show the effectiveness of our approach on five real datasets and two synthetic datasets. We make the following contributions:

- We introduce a principled formulation of a practical version of the retraining problem. We also provide its connection to existing formulations and offline reinforcement learning.

- We establish upper limits on the optimal number of retrains based on performance bounds and show how existing results can be used to determine whether you should retrain or not.

- We propose a novel retraining decision procedure based on performance forecasting. Our proposed algorithm is robust and outperforms existing baselines. It requires minimal performance data by fully leveraging the problem structure, employing compact regression models, and balancing the uncertainty caused by data scarcity through an uncertainty-informed decision process.

- We show that accounting for uncertainty in our method improves the performance.

## 2 RELATED WORK

We discuss related work and fields relevant to the retraining problem. A more detailed literature review, including connections to other related fields is provided in Appendix 8.1.

**Retraining problem** Few works explicitly target the retraining problem. Žliobaitė et al. (2015) propose a return on investment (ROI) framework to monitor and assess the retraining decision process, but do not introduce a method for actually deciding when to retrain. Mahadevan & Mathioudakis (2024) develop a retraining decision algorithm, CARA, which integrates the cost of retraining and introduces a "staleness cost" for persisting with an old model. CARA approximates the staleness

cost using offline data consisting of several trained models and their historical performance. Three versions of CARA are proposed: (i) retraining if the estimated staleness exceeds a threshold; (ii) retraining based on estimated cumulative staleness; or (iii) identifying an optimal retraining frequency. While providing promising results, CARA requires access to some of the data that will be used for retraining, and is very computationally intensive, so there is no adaptation to data obtained during the online decision period.

**Distribution shift detection** The retraining problem is closely connected to distribution shift detection and mitigation (Wang et al., 2024a; Hendrycks & Gimpel, 2017; Rabanser et al., 2019). Some approaches decide to adapt a model after detection of a changed distribution (Sugiyama & Kawanabe, 2012; Zhang et al., 2023). Since the signal is designed to adapt a model rather than trigger a full retraining, these methods are not appropriate as retraining signals. Other approaches, however, directly treat the detection of a distribution shift as a cue for retraining. ADWIN (Bifet & Gavaldà, 2007) uses statistical testing of the label or feature distribution. Another approach is to directly monitor the model's performance. FHDDM (Pesaranghader & Viktor, 2016) employs Hoeffding's inequality, while (Raab et al., 2020) relies on a Kolmogorov-Smirnov Windowing test. These approaches work well with low retraining costs, but perform poorly when retraining costs are high, as they tend to recommend retraining far too often. Additionally, they lack adaptability to varying costs, and it is difficult to determine the correct significance level to use for a given retraining-to-performance cost ratio.

**Offline reinforcement learning** Lastly, we discuss the connection to the offline reinforcement learning (ORL) setting, where the agent must learn a policy from a fixed dataset of rewards, actions, and states. This subset of RL is particularly challenging, as the agent cannot explore and can only rely on the dataset to infer the underlying dynamics and handle distribution shifts. See (Levine et al., 2020) for an extensive review. Q-learning and value function methods, which focus on predicting future action costs, have become the preferred approaches for ORL (Levine et al., 2020; Kalashnikov et al.; Hejna et al., 2023; Kostrikov et al., 2022). Some methods incorporate epistemic uncertainty into the Q-function to address distribution shifts of unseen actions (Kumar et al., 2020; Luis et al., 2023).

If we view the states as encoding both time and the model in use, and actions as either retraining or maintaining the current model, we can frame our problem as ORL. However, most existing RL approaches focus on scaling to large state or action spaces, employ large models, and assume access to abundant data, making them unsuitable for our context. A more detailed discussion on the connections and limitations of ORL methods is included in Appendix 8.9.

## 3 PROBLEM SETTING

In this section, we outline our formulation of the retraining problem. We have access to a sequence of datasets, $\mathcal{D}_{-w}, \ldots, \mathcal{D}_0, \ldots \mathcal{D}_T$ with features and labels $x_{i,t} \sim X_t, y_{i,t} \sim Y_t, \mathcal{D}_t = \{(x_{i,t}, y_{i,t})\}_{i=1}^{|\mathcal{D}_t|}$ , which are assumed to be drawn from a sequence of distributions $\mathcal{D}_t \sim p_t$ . In practice, this reflects the gradual distribution shifts that occur when collecting data over time, so we specifically cannot assume that $p_t = p_{t+1}$ (this would correspond to a special case of the problem, which we refer to as the no distribution shift case). The datasets are acquired at discrete times $t = [-w, \ldots, 0, \ldots, T]$. The sequence is split into an offline period that spans $t = [-w, \ldots 0]$, followed by an online period $[t = 1, \ldots T]$. At each time step $t$ of the online period, we are given the option to (re)train a model $f_t$, using the data acquired up until time $t$, for a retraining cost of $c_t$. The datasets and trained models can be formed and obtained through any means depending on the task at hand; for example, $f_1$ could be fine-tuned from $f_0$ and $\mathcal{D}_1$ could contain $\mathcal{D}_0$.

The complete sequence of decisions that we make can be encoded as a binary vector $\boldsymbol{\theta} \in \{0,1\}^T$, where $\theta_t = 1$ indicates that we retrain the model at time $t$. We introduce $r_{\boldsymbol{\theta}}(t)$ as a mapping function that returns the last training time at time $t$: $(r_{\boldsymbol{\theta}}(t) = \max_{t' \in \{0,t\} s.t. \boldsymbol{\theta}_{t'}=1} t'$, or $r_{\boldsymbol{\theta}}(t) = 0$ if $\|\boldsymbol{\theta}\|_1 = 0$.).

At each time step $t$, we are required to generate a certain number of predictions $N_t$ on a test set, which incurs a loss $\ell(\hat{y}, y)$, scaled by a cost $e_t$. This would correspond to actually using the model to make predictions, for example, to detect fraud – failing to detect a fraudulent transaction costs $e_t$, and approximately $N_t$ transactions are verified at time $t$. To make these predictions at time $t$, we use the most recently trained model, which we denote by $f_{r_{\boldsymbol{\theta}}(t)}$. To ensure that there is always at least one model available during the online period, we always train the last offline model $f_0$.

The target cost is a function of the vector $\boldsymbol{\theta}$, which encodes the retraining decisions, and combines the two opposing costs: the cost associated with model performance, $\sum_{t=1}^{T} e_t \sum_{i=1}^{N_t} \ell\big(f_{r_{\boldsymbol{\theta}}(t)}(x_{i,t}), y_{i,t}\big)$, and the cost to retrain, $\theta_t c_t$:

$$C_\alpha(\boldsymbol{\theta}) = \mathbb{E}\left[ \sum_{t=1}^{T} e_t \sum_{i=1}^{N_t} \ell\big(f_{r_{\boldsymbol{\theta}}(t)}(x_{i,t}), y_{i,t}\big) + \theta_t c_t \right]. \tag{1}$$

To make the expression more concise, we condense the expected loss into a scalar $pe_{i,j}$ where the two indices denotes the model index, and the timestep, respecvietly:

$$pe_{i,j} = \begin{cases} \mathbb{E}_{D_j}[\ell\big(f_i(X_j), Y_j\big)], & \text{if } i \le j, \\ 0, & \text{otherwise}. \end{cases} \tag{2}$$

We can simplify the problem by assuming a fixed cost of retraining, $c_t = c$, cost of loss, $e_t = e$, and number of predictions, $N_t = N$. The solutions we develop later in the paper are easily extended to the case where these are varying, but known, quantities. Introducing the cost-to-performance ratio parameter $\alpha = \frac{c}{eN}$, the online objective can be compactly written as:

$$C_\alpha(\boldsymbol{\theta}) = eN\left( \alpha\|\boldsymbol{\theta}\|_1 + \sum_{t=1}^{T} pe_{r_{\boldsymbol{\theta}}(t),t} \right). \tag{3}$$

## 3.1 OFFLINE AND ONLINE DATA

The cost $C_\alpha(\boldsymbol{\theta})$ is only evaluated over the online period. We assume that we have access to all the datasets and trained models during the offline period. In practice, the number of models and datasets is typically limited to only a few (around 10 to 20 at most), which is why we characterize this problem as being in a low-data regime. We denote this data as $\mathcal{I}^{offline} = (\mathcal{D}_{-w}, \ldots \mathcal{D}_0, f_{-w}, \ldots, f_0)$. In the online mode, each decision at time $t$ can only rely on information available prior to that time, which we denote by $\mathcal{I}_{<t}$. $\mathcal{I}_{<t}$ therefore contains both the offline data $\mathcal{I}^{offline}$, and the online data that was collected up to the timestep $t$: $\mathcal{I}_{<t}^{online}$. The online data is similar to the offline data, but it only contains the models that were actually trained; $\mathcal{I}_{<t}^{online} = (\mathcal{D}_1, \ldots \mathcal{D}_{t-1}, \{f_i\}_{i \text{ s.t. } \theta_i=1}))$.

Each entry of $\boldsymbol{\theta}$ can therefore be modeled by a binary function $g(t, \mathcal{I}_{<t}) \in \{0, 1\}$:

$$\boldsymbol{\theta} = [g(1, \mathcal{I}^{offline}), \ldots, g(T, \mathcal{I}_{<T)})]^\top. \tag{4}$$

Given $c_t$, $e_t$, and $N_t$, the task is to determine the $g$ that generates the retraining schedule $\boldsymbol{\theta}^*$ that minimizes the cost $C_\alpha(\boldsymbol{\theta})$;

## 3.2 SOME ANALYSIS

$$\boldsymbol{\theta}^* = \underset{\boldsymbol{\theta} \in \{0,1\}^T}{\arg\min}\, C_\alpha(\boldsymbol{\theta}). \tag{5}$$

Before introducing methods that learn to generate such a schedule $\boldsymbol{\theta}$, we begin by providing some basic properties of the problem. Specifically, we establish bounds on the number of retraining actions of the optimal solution. These can be used to determine whether we even need to consider retraining. We also provide guidance on leveraging existing performance bounds (such as scaling laws) to compute the relevant quantities in these bounds. These theoretical insights can be used to derive a practical rule of thumb on a case-by-case basis.

Our upper bound mainly depends on the difference between the expected performance of a model trained on dataset $\mathcal{D}_i$ and the performance of a model trained on the subsequent dataset $\mathcal{D}_{i+1}$, evaluated on the same dataset from any timestep $\mathcal{D}_t$ :

$$L \ge |pe_{i,t} - pe_{i+1,t}| \, \forall t \in [T] \tag{6}$$

Given this quantity, we derive the following result of an upper bound for the number of retrains of the optimal solution, which we denote by $r^* = \|\boldsymbol{\theta}^*\|_1$:

**Proposition 3.1.** *Given that $L \ge |pe_{i,t} - pe_{i+1,t}| \, \forall t \in [T]$, a horizon of $T \in \mathbb{N}$, and a relative cost of retrain $\alpha$, the number of retrains of the solution to Equation 5 $r^* \triangleq \|\boldsymbol{\theta}^*\|_1$ satisfies:*

$$r^* \le T - \sqrt{\frac{\alpha}{L}} \tag{7}$$

The proof is provided in Appendix 8.2. Suppose a practitioner has reasonable approximations of $L$ and $\alpha$, and a horizon to consider, $T$. Then if $T - \sqrt{\left(\frac{\alpha}{L}\right)} < 1$, no retraining should be performed. We demonstrate how this result should be used in practice in Appendix 8.2.1.

**Bounding** $L$     General bounds for $L$ are too loose to be helpful; however, in some cases, reasonable estimates can be derived. For the specific cases of "no distribution shift" IID data, where the data simply accumulates ($\mathcal{D}_t \subset \mathcal{D}_{t+1}, \mathcal{D}_t \sim p(\mathcal{D}) \forall t$), we can leverage some known theoretical result, such as Probably Approximately Correct (PAC) learning theory (Valiant, 1984) or Rademacher Complexity (Bartlett & Mendelson, 2002). Even in real-world applications, where data often exhibit temporal or spatial dependencies, making the non-distribution shift IID assumption unrealistic, bounds have been derived using stability analysis (Mohri & Rostamizadeh, 2007; 2010) or tailored Rademacher complexity bounds (Mohri & Rostamizadeh, 2008). For large-scale training settings, precise empirical scaling laws have been derived (Kaplan et al., 2020; Hoffmann et al., 2024). Kaplan et al. (2020) derive that the loss $\mathcal{L}$ of the neural network scales with respect to the dataset size $N$ as $\mathcal{L} = \left(N/5.4 \cdot 10^{13}\right)^{-0.095}$. Such scaling laws enable the accurate estimation of expected performance improvements from expanded datasets $L$. Thus, they enable informed decisions about when retraining would yield substantial benefits. For a more detailed discussion see Appendix 8.3.

## 4    METHODOLOGY

A retraining decision algorithm must specify the decision functions $g_\phi(t, \mathcal{I}_{<t}) \in \{0, 1\}$ (where $\phi$ contains the parameters of the algorithm) used to build the decision vector $\boldsymbol{\theta}$. To make perfect decisions, we would need future performance values, i.e., $pe_{i,j} \forall (i > t$ or $j > t)$. This is infeasible; however, we assume that there is an underlying temporal autocorrelation between the performance of different models trained at different times, which we aim to exploit to build a predictive model. We therefore propose to 1) model these future values as random variables and learn their distributions; and 2) base our decisions on the predicted distributions to construct our method, the Uncertainty-Performance Forecaster (UPF). As our methodology involves forecasting future performance as a key subtask, we evaluate and quantify the impact of success in this task on the overall performance of our algorithm, as detailed in Appendix 8.5.

### 4.1    PERFORMANCE FORECASTER

The first component of our algorithm involves learning a performance predictor to forecast unknown entries in $pe$, which are defined as $pe_{i,j} = \mathbb{E}_{D_j}[\ell(f_i(X_j), Y_j)]$ for $i \leq j$ (see Eqn 2). In a classification setting where we consider the 0-1 loss $\ell(y', y) = \mathbb{1}[y' \neq y]$, these are $1 - accuracy$. We introduce random variables $A_{ij}$ and model the entries $pe_{ij}$ as realizations of these.

Since the $A_{i,j}$ random variables are bounded, we model them (after appropriate scaling) as Beta distributed with parameters $\alpha(\mathbf{r}_{i,j}), \beta(\mathbf{r}_{i,j})$ that depend on some input feature $\mathbf{r}_{i,j}$. We also define their associated mean $\mu(\mathbf{r}_{i,j})$ and variance $\sigma(\mathbf{r}_{i,j})$. Given the parameters $\alpha(\mathbf{r}_{i,j}), \beta(\mathbf{r}_{i,j})$, we model the random variables to be independent of each other:

$$P\left(A_{0,0}, \ldots, A_{T,T} | \{\alpha(\mathbf{r}_{i,j}), \beta(\mathbf{r}_{i,j})\}_{i \leq j}^T\right) = \prod_{i \leq j} P(A_{i,j} | \alpha(\mathbf{r}_{i,j}), \beta(\mathbf{r}_{i,j})), \tag{8}$$

$$= \prod_{i \leq j} \text{Beta}(\alpha(\mathbf{r}_{i,j}), \beta(\mathbf{r}_{i,j})). \tag{9}$$

where Beta() denotes the pdf of a Beta distribution. We choose the input features $\mathbf{r}_{ij}$ to include the indices of the training and evaluation datasets ($i$ and $j$, respectively), along with additional features that capture the gap between the training and evaluation timesteps (the difference $j - i$, and summary statistics of the distribution shift $z_{shift}$ (see Appendix 8.5 for details). The input features are thus given by $\mathbf{r}_{i,j} = [i, j, j - i, z_{shift}]$.

From the offline data, we have access to observations $a_{i,j} \sim A_{i,j}$, and can build a regression dataset to learn the parameters $\alpha(\mathbf{r}_{i,j}), \beta(\mathbf{r}_{i,j})$. We specify the learning task by constructing $(\mathbf{r}_{i,j}, a_{i,j})$ pairs:

$$\mathcal{M}_{<t} = \{(\mathbf{r}_{i,j}, a_{i,j}); \forall f_i \in \mathcal{I}_{<t}, \forall \mathcal{D}_j \in \mathcal{I}_{<t}\}. \tag{10}$$

Direct learning of the $\alpha, \beta$ parameters can be unstable. Therefore, we use a Gaussian approximation:

$$\text{Beta}(\alpha(\mathbf{r}_{i,j}), \beta(\mathbf{r}_{i,j})) \approx \mathcal{N}(\mu(\mathbf{r}_{i,j}), \sigma(\mathbf{r}_{i,j})), \tag{11}$$

This allows use to write the likelihood of our dataset as:

$$\mathcal{L}(\mathcal{M}_{<t}; \phi) = \prod_{i,j \in \mathcal{M}_{<t}} P(a_{i,j} | \mathbf{r}_{i,j}, \phi) = \prod_{i,j \in \mathcal{M}_{<t}} \mathcal{N}(a_{i,j}; \mu_\phi(\mathbf{r}_{i,j}), \sigma_\phi(\mathbf{r}_{i,j})). \tag{12}$$

We parameterize the variance as a constant $\sigma_\phi(\mathbf{r}_{i,j}) = \sigma_\phi$. Maximizing the likelihood w.r.t. to the mean parameters $\mu_\phi(\mathbf{r}_{i,j})$ then becomes a standard mean square error objective. Given the expectation of operating in a very low-data regime, we rely on simple inference models, such as linear regression. Once these parameters are learned, we can recover the corresponding $\alpha_\phi(\mathbf{r}_{i,j}), \beta_\phi(\mathbf{r}_{i,j})$ parameters to obtain our predictive distribution (see Appendix 8.5 for additional details) ;

$$P_\phi(A_{i,j}) = Beta(\alpha_\phi(\mathbf{r}_{i,j}), \beta_\phi(\mathbf{r}_{i,j})). \tag{13}$$

As stated, this parameterization is appropriate for bounded losses. Other distributions can be used to model different loss domains if needed as we show in Appendix 8.6. As $\mathcal{I}_{<t}$ grows at each time step, our training data increases, so we retrain and obtain a new $P_\phi(A_{i,j})$ each time.

## 4.2 DECISIONS UNDER UNCERTAINTY

Now we describe how we use $P_\phi(A_{i,j})$ to decide whether to retrain. We introduce a random variable $\tilde{C}$ that represents the total cost (Eqn. 3) (given a sequence of decisions $\boldsymbol{\theta}$):

$$\tilde{C}(\boldsymbol{\theta}) = eN\left(\alpha\|\boldsymbol{\theta}\|_1 + \sum_{t=1}^{T} A_{r_{\boldsymbol{\theta}}(t),t}\right). \tag{14}$$

We can therefore define our decision rule based on this random cost using our learned distribution of performances $P_\phi(\bar{A}_{i,j})$. Given the past decisions $\boldsymbol{\theta}_{<t}$, our next decision $\tilde{\boldsymbol{\theta}}_t$ is obtained by comparing the $\delta$-level quantiles of the total cost incurred if we retrain, denoted by $\tilde{C}_{\boldsymbol{\theta}_{<t}}|retrain$, and the cost incurred if we do not, denoted by $\tilde{C}_{\boldsymbol{\theta}_{<t}}|keep$. Using $F_X^{-1}(\delta)$ as the quantile function of a random variable, our rule is given by:

$$\tilde{\boldsymbol{\theta}}_t = \mathbb{1}\left[F_{\tilde{C}_{\boldsymbol{\theta}_{<t}}|retrain}^{-1}(\delta) < F_{\tilde{C}_{\boldsymbol{\theta}_{<t}}|keep}^{-1}(\delta)\right]. \tag{15}$$

The quantile parameter $\delta$ allows us to control how conservative we are. Lower values of $\delta$ lead to decisions that prioritize costs with lower variance, while setting $\delta = 0.5$ simply selects the decision that minimizes the expected total cost. As defined, the retraining decision $\tilde{\boldsymbol{\theta}}_t$ is deterministic.

We begin by giving explicit expressions for the conditional random variables $\tilde{C}_{\boldsymbol{\theta}_{<t}}|retrain$ and $\tilde{C}_{\boldsymbol{\theta}_{<t}}|keep$. If we decide to retrain at time step $t$, the incurred costs include the retraining cost $\alpha$, the performance cost of the most recent model $A_{t,t}$, and future costs for the decisions we will make. Specifically, we incur $\tilde{C}_{\boldsymbol{\theta}_{<t+1}}|retrain$ if the next decision is to retrain, and $\tilde{C}_{\boldsymbol{\theta}_{<t+1}}|keep$ if it is not. If we choose not to retrain and keep the current model, we only incur the performance cost of the old model, $A_{r_{\boldsymbol{\theta}}(t-1),t}$.

These random variables can therefore be recursively defined as follows:

$$\tilde{C}_{\boldsymbol{\theta}_{<t}}|retrain = \alpha + A_{t,t} + \tilde{\boldsymbol{\theta}}_{t+1}\tilde{C}_{\boldsymbol{\theta}_{<t+1}}|retrain + (1 - \tilde{\boldsymbol{\theta}}_{t+1})\tilde{C}_{\boldsymbol{\theta}_{<t+1}}|keep \tag{16}$$

$$= \alpha + A_{t,t} + \sum_{t'=t+1}^{T} A_{r_{\tilde{\boldsymbol{\theta}}}(t'),t'} + \alpha\tilde{\boldsymbol{\theta}}_{t'} \tag{17}$$

$$\tilde{C}_{\boldsymbol{\theta}_{<t}}|keep = A_{r_{\boldsymbol{\theta}}(t-1),t} + \sum_{t'=t+1}^{T} A_{r_{\tilde{\boldsymbol{\theta}}}(t'),t'} + \alpha\tilde{\boldsymbol{\theta}}_{t'} \tag{18}$$

As shown, the cost random variables are constructed recursively by summing the distribution of the cost of performances $A_{i,j}$ that would be selected by the decision rule $\tilde{\boldsymbol{\theta}}$, as $\tilde{\boldsymbol{\theta}}$ and the $\alpha$ parameter are both deterministic.

The decision rule introduced in Eqn. 15 can therefore be written as:

$$\tilde{\boldsymbol{\theta}}_t = \mathbb{1}\left[F_{\alpha+A_{t,t}+\sum_{t'=t+1}^{T} A_{r_{\tilde{\boldsymbol{\theta}}}(t'),t'}+\alpha\tilde{\boldsymbol{\theta}}_{t'}}^{-1}(\delta) < F_{A_{r_{\boldsymbol{\theta}}(t-1),t}+\sum_{t'=t+1}^{T} A_{r_{\tilde{\boldsymbol{\theta}}}(t'),t'}+\alpha\tilde{\boldsymbol{\theta}}_{t'}}^{-1}(\delta)\right]. \tag{19}$$

We use the learned Beta distributions, introduced in the previous section, plugging them into Eqn. 19 in order to make a retraining decision.

If the parameterization $P_\phi(A_{i,j})$ does not lead to a closed form expression, we use Monte Carlo methods to obtain quantile estimates:

$$F^{-1}_{C_{\boldsymbol{\theta}_{<t}}|retrain}(\delta) \approx \hat{F}^{-1}_{C_{\boldsymbol{\theta}_{<t}}|retrain}(\delta) \tag{20}$$

where $\hat{F}^{-1}_{C_{\boldsymbol{\theta}_{<t}}|retrain}(\delta)$ is obtained through bootstrapping.

**Connection to offline reinforcement learning** The formulation closely resembles a $Q$-learning formulation. The $C$ values defined in Eqns. 16- 18 strongly align with $Q$ functions. Indeed, one possible approach is to bypass the learning of the $pe$ and directly optimize the decision-making process using $Q$-learning approaches. The problem we are considering can be viewed as a corner case of offline RL, where the state space is finite and enumerable, the training data are extremely limited, the transition function is deterministic and fully known, and the reward structure is highly structured. In fact, our methodology can be reinterpreted as an offline variant of a $Q$-learning approach with a specific parameterization of the $Q$ function, further justifying the motivation behind our method. We explore and formalize this connection in Appendix 8.9. However, as we have explained in the related work section, existing ORL methods are not suitable for this setting. We provide the results for one ORL baseline in Appendix 8.9 to examplify that point.

# 5 EXPERIMENTS

**Evaluation Metrics** The performance of a retraining decision method is evaluated based on both the average performance and the total retraining cost. The tradeoff between these factors is controlled by $\alpha$. When using the zero-one loss in classification, $\alpha$ can be seen as the ratio of retraining cost to the cost of misclassifications. In practice, $\alpha$ is application-dependent and should be set by the practitioner. The retraining cost would be low (small $\alpha$) for situations such as fine-tuning small models. By contrast, when retraining large language models, or in high-stakes settings requiring extensive validation, the retraining cost is high (large $\alpha$). The retraining decision method should be robust across all scenarios. The appropriate value of $\alpha$ can be very difficult to estimate and will likely be an approximation in practice. Consequently, we present experiments that test the robustness of the method to inaccuracies in $\alpha$ in Section 6.

In our experiments, we address classification tasks with a zero-one loss, and set $eN = 1$. We report an empirical estimate of the target cost $\hat{C}_\alpha(\boldsymbol{\theta})$ (Eqn. 3), obtained from the test set, over varying $\alpha$:

$$C_\alpha(\boldsymbol{\theta}) \approx \hat{C}_\alpha(\boldsymbol{\theta}) \triangleq \alpha\|\boldsymbol{\theta}\|_1 + \sum_{t=1}^{T} pe^{test}_{r_{\boldsymbol{\theta}}(t),t}, \tag{21}$$

where $pe^{test}_{i,j} = 1 - acc^{test}$ with $\ell(y,y') = \mathbb{1}[y \neq y']$, To summarize the results at multiple $\alpha$ operating points, we report the area-under-the-curve (AUC) of $\hat{C}_\alpha(\boldsymbol{\theta})$. We compute 10 $\alpha$ operating points and we allow $\alpha$ to range from 0 (no retrain cost) to $\alpha_{max}$ (where the cost is too high to justify any retraining). The upper bound, $\alpha_{max}$, is determined by the $\alpha$ value at which the oracle reaches 0 retrains.[1] The oracle is obtained by determining the optimal schedule that minimizes the target cost, assuming exact knowledge of all future $pe_{ij}$ entries, i.e., $\boldsymbol{\theta}^{oracle} = \arg\min_{\boldsymbol{\theta}} \hat{C}_\alpha(\boldsymbol{\theta})$.

**Datasets** We present results on synthetic and real datasets. For the real datasets, we use datasets with a timestamp for each sample and partition the data in time to create a sequence of datasets $\mathcal{D}_0, \mathcal{D}_1, \ldots$. For each trial, we sample a different sequence of length $w + T$ within the complete dataset sequence available. We report results on: (i) the **electricity** dataset (Harries et al.), a binary classification task predicting the rise or fall of electricity prices in New South Wales, Australia; (ii) the **airplane** dataset (Gomes et al., 2017), which records whether a flight is delayed; (iii) **yelpCHI** (Dou et al., 2020), which classifies if a user's review is legitimate; and (iv) **epicgames** (Ozmen et al., 2024), where the task is to predict whether an author's critique of a game was selected as a top critique. As a base model $f$, we use XGBoost (Chen & Guestrin, 2016).

---

[1]The use of the oracle to define the range of $\alpha$ values for the AUC computation does not bias the performance assessment via pollution with future knowledge. None of the algorithms makes use of the oracle information. Using the oracle merely ensures that the performance comparison is conducted over the range of relevant $\alpha$.

Table 1: AUC of the combined performance/retraining cost metric $\hat{C}_\alpha(\boldsymbol{\theta})$, computed over a range of $\alpha$ values, for all datasets. The bolded entries represent the best, and the underlined entries indicate the second best. The $*$ denotes statistically significant difference with respect to the next best baseline, evaluated using a Wilcoxon test at the $5\%$ significance level.

| | electricity | Gauss | circles | airplanes | yelpCHI | epicgames | iWild |
|---|---|---|---|---|---|---|---|
| ADWIN-5% | 2.8099 | 0.4533 | 0.0753 | 2.6353 | 0.1298 | 0.3217 | 3.7371 |
| ADWIN-50% | 2.8131 | 0.4848 | 0.0753 | 2.7147 | 0.1298 | 0.3238 | 4.2564 |
| KSWIN-5% | 3.8979 | 0.3975 | 0.0753 | 3.2300 | 0.1322 | 0.3420 | 4.4268 |
| KSWIN-50% | 4.0521 | 0.9530 | 0.0794 | 3.2042 | 0.1655 | 0.3537 | 4.4268 |
| FHDDM-5% | 3.1525 | 0.3893 | 0.0753 | 2.6577 | 0.1324 | 0.3298 | 4.4267 |
| FHDDM-50% | 3.4037 | 0.5918 | 0.0772 | 2.7077 | 0.1450 | 0.3389 | 4.4268 |
| CARA cumul. | 2.7147 | 0.3862 | 0.0731 | 2.2900 | 0.1299 | 0.3228 | 3.8922 |
| CARA per. | 2.8986 | 0.4678 | 0.0800 | 2.4061 | 0.1318 | 0.3260 | 3.7527 |
| CARA | 2.7198 | 0.3841 | 0.0726 | **2.2753*** | 0.1294 | 0.3202 | 3.9506 |
| UPF (ours) | **2.5782*** | **0.3829*** | **0.0668*** | 2.2865 | **0.1293*** | **0.3189*** | **3.0498*** |
| oracle | 2.4217 | 0.3724 | 0.0627 | 2.2298 | 0.1275 | 0.3170 | 2.4973 |

We also present a larger vision dataset that requires a larger network to process. **iWildCam** (Beery et al., 2020) consists of images of animals in the wilderness, captured at various locations, and the task involves multi-class animals classification. Our approach utilizes a pretrained vision model, augmented with a linear layer that processes the image representation along with the location domain to produce the final classification output. We allow for a different pretrained architecture model at each timestep $t$, and perform a random search over a set of 188 choices from the Huggingface library (Wightman, 2019). These encompass a wide variety of networks, including ViT (Dosovitskiy et al., 2021), ResNet (He et al., 2015) and convolution based (O'Shea & Nash, 2015). Appendix 8.4 provides additional details on the architecture, training procedure, and hyperparameter search. For the synthetic dataset, we follow Mahadevan & Mathioudakis (2024) to generate two 2D datasets with covariate shift (**Gauss**) and concept drift (**circles**) (Pesaranghader et al., 2016). Appendix 8.4 contains details on the generation. We report 3 trials for **iWild** and 10 trials for the other datasets.

**Baselines and algorithm settings** We set the confidence threshold of our UPF algorithm to $\delta = 95\%$, as it is a standard value used for confidence intervals. For $\mu_\phi(\mathbf{r}_{i,j})$, we use a linear regression model, ElasticNetCV (Zou & Hastie, 2005), from the scikit-learn library. All other optimization parameters are set to default choices from the scikit learn libraries. We report results on shift detection baselines and the three variants of the CARA baseline, as well as the **oracle**.

For the **distribution shift detection** baselines, we set the window size to the size of an individual dataset $|\mathcal{D}|$, and retrain when the algorithm detects a distribution shift. (Then we reset the algorithm with the dataset of the last retrained model.) As these methods cannot take into account the cost of retraining, we vary the significance level threshold $\delta$ to obtain different frequencies of retraining. We include **ADWIN-**$\delta$ (Bifet & Gavaldà, 2007), which is based on statistical testing of the label distribution, **FHDDM-**$\delta$ (Pesaranghader & Viktor, 2016), which is based on Hoeffding's inequality, and **KSWIN-**$\delta$ (Raab et al., 2020), which is based on the Kolmogorov-Smirnov test.

**CARA** (Mahadevan & Mathioudakis, 2024) searches for the best strategy with fixed parameters using the offline data. The standard version, **CARA**, searches for the best threshold of approximate performance and retrains when it drops below it. The cumulative version, **CARA cumul.**, searches for the best threshold of the cumulated approximate performance; and the periodic strategy, **CARA per.**, searches for the best retraining frequency. Appendix 8.10 provides additional details on the CARA baseline in the context of our experiments.

## 6 RESULTS

We start by presenting in Table 1 the area-under-the-curve (AUC) of the total cost value $\hat{C}_\alpha(\boldsymbol{\theta})$. The AUC is computed as the area over a range of $\alpha$ values determined by the oracle performance. Lower values of AUC are better because we aim to reduce the cost over the operating range. Overall, we

Table 2: We compare the best performing algorithms for the electricity dataset with the optimal decisions (the oracle) in both high and low retraining cost settings. For each baseline, we report the number of retrains and the average accuracy, as well as our primary metric $\hat{C}_\alpha(\boldsymbol{\theta})$ that combines both factors using $\alpha$. The results show that the proposed method achieves the best $\hat{C}_\alpha(\boldsymbol{\theta})$ value and closely approximates the oracle's behavior in both scenarios, highlighted in bold.

| | High retrain cost $\alpha$ = 0.9 | | | Low retrain cost $\alpha$ = 0.1 | | |
|---|---|---|---|---|---|---|
| | #retrain | Average Acc | $\hat{C}_\alpha(\boldsymbol{\theta})$ | #retrain | Average Acc | $\hat{C}_\alpha(\boldsymbol{\theta})$ |
| ADWIN-5% | 1.0 ± 0.58 | 0.7 ± 0.04 | 3.27 ± 0.4 | 1.0 ± 0.58 | 0.7 ± 0.04 | 2.47 ± 0.25 |
| ADWIN-50% | 1.17 ± 0.38 | 0.72 ± 0.03 | 3.32 ± 0.32 | **1.17 ± 0.38** | 0.72 ± 0.03 | 2.39 ± 0.26 |
| CARA | **0.0 ± 0.0** | 0.65 ± 0.02 | **2.78 ± 0.19** | 0.33 ± 0.75 | 0.66 ± 0.04 | 2.73 ± 0.25 |
| CARA cumul. | **0.0 ± 0.0** | 0.65 ± 0.02 | **2.78 ± 0.19** | 0.33 ± 0.48 | 0.67 ± 0.02 | 2.68 ± 0.18 |
| CARA per. | 1.0 ± 0.0 | 0.69 ± 0.02 | 3.34 ± 0.14 | 1.0 ± 0.0 | 0.69 ± 0.02 | 2.54 ± 0.14 |
| UPF (ours) | **0.1 ± 0.3** | 0.68 ± 0.04 | **2.69 ± 0.26** | **2.5 ± 0.67** | 0.75 ± 0.03 | **2.24 ± 0.17** |
| oracle | 0.0 ± 0.0 | 0.66 ± 0.03 | 2.68 ± 0.26 | 5.6 ± 1.44 | 0.83 ± 0.02 | 1.93 ± 0.06 |

see that our proposed method achieves the best trade-off between the number of retrains and average accuracy across all baselines and datasets. To gain better insight into the behavior of the different algorithms and how they are impacted by varying retraining cost parameters, we provide a detailed overview for one dataset with two values of $\alpha$: one where the cost of retraining is low and one where it is high, as shown in Table 2. Figure 2 depicts how the the total cost $\hat{C}_\alpha(\boldsymbol{\theta})$ and the number of retrains vary as $\alpha$ is changed. Appendix 8.8 contains the complete set of results and figures. First, examining the behavior of the optimal solution (**oracle**), we unsurprisingly observe that in the high retraining cost scenario, both the number of retrains and the average accuracy are lower, while in the low retraining cost scenario, the number of retrains and the average accuracy are higher.

Next, we observe that the proposed UPF method follows the oracle more closely than the other baselines and is more sensitive to the $\alpha$ parameter compared to the cost-aware method (CARA). This is particularly apparent in Figure 2. The CARA baselines relies heavily on its assumptions about performance and is therefore not as robust in scenarios where those assumptions do not hold. The detection shift methods cannot take the varying parameters as input, so the results remain the same for both values of $\alpha$. Since these methods do not account for retraining costs, they perform better when the cost is very low, as they simply retrain whenever a shift is detected. This can be a good strategy if retraining costs little. Indeed, we observe that all ADWIN and FHDDM variants are closer to the optimal values in the low range of $\alpha$ in the left of Figure 2. However, as the cost of retraining increases, these methods become impractical. Varying the threshold can yield better results—a lower significance requirement (50%) allows for more retraining and therefore works better when retraining costs are low, while the inverse holds in a high-cost regime, where a more conservative retraining strategy is preferable. However, it is not possible to know in advance which significance threshold should be used for a given $\alpha$, making these methods largely impractical for such a setting.

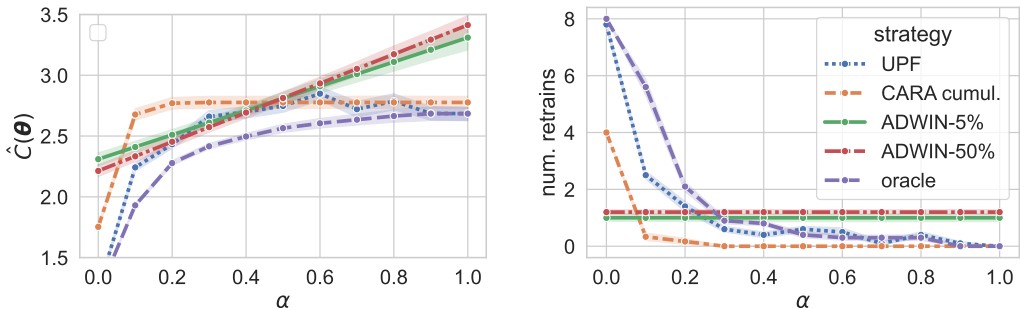

Figure 2: Results on the electricity dataset. **Left)** Cost $\hat{C}_\alpha(\boldsymbol{\theta})$ vs $\alpha$. **Right)** Number of retrains vs $\alpha$. In the left figure, we can see that UPF consistently reaches low $\hat{C}_\alpha(\boldsymbol{\theta})$ across different $\alpha$. In the right figure, the number of retrainings of UPF follows the optimal baseline more closely.

Table 3: Ablation study on accounting for uncertainty in our prediction. Targeting the 95% quantile is better overall than the deterministic approach (equivalent to a 50% quantile). The $*$ denotes statistically significant difference with respect to the next best baseline, evaluated using a Wilcoxon test at the 5% significance level.

|     | electricity | gauss | circles | airplanes | yelp | epicgames |
|-----|-------------|-------|---------|-----------|------|-----------|
| PF  | **2.5884 ± 0.13**\* | 0.3673 ± 0.03 | 0.0697 ± 0.01 | 2.3688 ± 0.35 | 0.1180 ± 0.00 | 0.3211 ± 0.01 |
| UPF | 2.6056 ± 0.14 | **0.3643 ± 0.03**\* | **0.0670 ± 0.01**\* | **2.2688 ± 0.26**\* | **0.1175 ± 0.00**\* | **0.3202 ± 0.00**\* |

**Ablation study - The importance of uncertainty** In our approach, we model the distribution of future costs and set targets at the 95% quantile to ensure robustness against noisy predictions. To assess whether this strategy enhances robustness and improves performance, we compare the proposed UPF algorithm, with the 95% quantile, against a deterministic version, referred to as PF, which selects the predicted decision that minimizes costs. This corresponds to setting the quantile to 50% in our algorithm (PF = UPF-50%). We observe in Table 3 that relying on conservative quantiles in our predictions results in better overall outcomes, compared to the deterministic version, PF, with statistical significance observed across all datasets except for electricity.

**Robustness to wrong $\alpha$** In our setting, we assume that the relative cost of performance and re-training $\alpha$ is known. However, in practice, this tradeoff value can be hard to estimate accurately. It is therefore of high practical interest to assess the impact of a misspecified $\alpha$ value, and to identify the settings where misspecification is the most impactful. In Figure 3, we present how wrongly speci-fied $\alpha$ values impact the performance of our algorithm and the CARA baseline on one dataset. Both algorithms are reasonably robust, as it requires a large deviation from the true $\alpha$ value (upper right and bottom left) to start seeing a degradation of performance of more than 1%. UPF is generally more robust to changes of $\alpha$. Both algorithms are more susceptible to overestimation of $\alpha$.

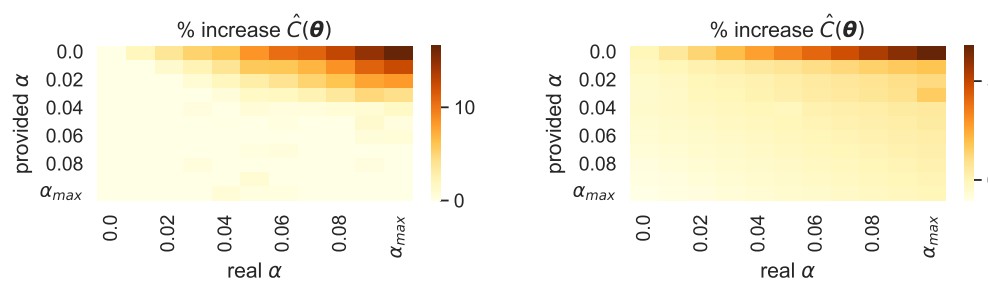

Figure 3: Impact of wrong $\alpha$ measured by the percentage increase of $\hat{C}_\alpha(\boldsymbol{\theta})$ on the epicgames dataset. **left)** CARA **right)** UPF. Overall, both methods are reasonably robust to a wrong $\alpha$ specifi-cation, with UPF being the more robust.

## 7 CONCLUSION AND LIMITATIONS

We have proposed a practical formulation of the important problem of model retraining, which has been neglected in the literature, and highlighted its complexity. Our method outlines a promis-ing avenue, as our experiments have shown that even with distribution shift, it is not unreasonable to expect some patterns in future performance that could be predicted with the help of uncertainty mod-eling. This data-driven approach is lightweight, practical, and outperforms existing approaches. It is robust to varying cost settings and has demonstrated resilience to misspecified cost-to-performance ratios. We have also highlighted the quantities of interest to estimate in order to better understand the characteristics of a specific problem. While our study demonstrates promising results in predicting optimal retraining schedules, several aspects warrant further exploration. Our main experiments in-vestigate a setting where the offline dataset ($w = 7$) is non-negligible in size. However, we achieved good performance even with a reduced dataset, which shows that initial training costs can be reduced (see Appendix 8.11).We evaluated the method individually for each dataset, but future work could further reduce costs by transferring schedulers across datasets and tasks. Additionally, adapting techniques from Hyperparameter Optimization could enhance performance forecasting.

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

# 8 APPENDIX

## 8.1 EXTENDED DISCUSSION OF RELATED WORK

**Retraining problem** Few works explicitly target the retraining problem. Žliobaitė et al. (2015) propose a return on investment (ROI) framework to monitor and assess the retraining decision process. Mahadevan & Mathioudakis (2024) develop a retraining decision algorithm, CARA, which integrates the cost of retraining into its formulation. It introduces the concept of a "staleness cost" which represents the cost of not retraining. The approach involves approximating the staleness cost and optimizing various strategies to reduce the overall cost, based on some offline data. The offline data consist of a few trained models, each with an associated dataset that was collected prior to the retraining decision process. Mahadevan & Mathioudakis (2024) propose three methods: the first retrains when the estimated staleness cost exceeds a threshold; the second tracks the accumulated

staleness cost and applies a threshold on that value; and the third searches for the optimal retraining frequency. The staleness cost approximation for using a model on a dataset relies on the loss of individual known samples. This loss is scaled by the average similarity between the features of these known samples and the features of the dataset of interest. Consequently, it assumes access to the features of some of the samples at a given time before deciding to retrain. Moreover, the search for the threshold or the period is computationally intensive and therefore can only be done once using some offline data; it cannot modify the parameters as new information arrives.

**Distribution shift detection** The retraining problem is closely connected to distribution shift detection and mitigation (Wang et al., 2024a; Hendrycks & Gimpel, 2017; Scheirer et al., 2013; Cerqueira et al., 2021; Bar-Shalom et al., 2023; Rabanser et al., 2019). Some methods adapt the model to adjust to evolving distributions (Sugiyama & Kawanabe, 2012; Zhang et al., 2023; Fang et al., 2020; Pesaranghader et al., 2018). Since the signal is designed to adapt a model rather than trigger a full retraining, these methods are not appropriate as retraining signals. Some approaches, however, directly treat the detection of a distribution shift as a cue for retraining (Bifet & Gavaldà, 2007; Pesaranghader & Viktor, 2016; Raab et al., 2020), and can be used as baselines. ADWIN (Bifet & Gavaldà, 2007) uses statistical testing of the label or feature distribution. Another approach is to directly monitor the model's performance. FHDDM (Pesaranghader & Viktor, 2016) employs Hoeffding's inequality, while (Raab et al., 2020) relies on a Kolmogorov-Smirnov Windowing test. These approaches may work well when retraining costs are low, but they become unsuitable when retraining is expensive – it is not always optimal to retrain after every minor shift. This is tied to a more general weakness of lacking adaptability to varying costs. While the significance level parameter can be adjusted, the appropriate significance level for a given retraining-to-performance cost ratio is unknown and difficult to estimate.

**Changepoint detection** Another closely related field is changepoint detection, which is similar to the distribution shift problem. Changepoint detection is the task of identifying points in a sequence where the statistical properties of the data change abruptly. This problem was introduced and presented by Adams & MacKay (2007), where they aim to infer the most probable distribution of the most recent changepoint in an online setting. Recent work, such as (Li et al., 2021), has expanded on this problem in ways closer to our retraining setting, as they incorporate adaptation into the changepoint detection process,The sensitivity of the detection is controlled by certain sensitivity parameters.

However, to transform the changepoint detection problem formulated by Li et al. (2021) into the retraining problem we consider, we would need to introduce a cost for adaptation, a cost for accuracy loss, and then formulate an optimization problem to find the appropriate sensitivity parameter for achieving the optimal number of adaptations. However, since this parameter lacks a specific physical or practical meaning, it is unclear beforehand how the choice of its value will impact the adaptation rate. Furthermore, in our setting, the optimal rate of adaptation (or retraining frequency) is unknown. Determining this optimal retraining frequency is one of the major challenges of the retraining problem.

**Bayesian Optimization** Our method is based on forecasting future model performance using historical data. This approach closely aligns with Bayesian Optimization (see (Shahriari et al., 2016) for a review on this topic), commonly used in the Hyperparameter Optimization (HPO) field. The Freeze-Thaw method, introduced by Swersky et al. (2014), leverages Gaussian Processes to predict the trajectory of validation loss, enabling early stopping and optimization of the hyperparameter search space. It remains a relevant technique (Rakotoarison et al., 2024). Similarly, Dai et al. (2019) derive a Bayes-optimal stopping rule using a related approach. This method can be extended to predict the performance of other models and address hyperparameter optimization challenges (Wang et al., 2024b). In our context, we predict the performance of different models under potential distribution shifts, but the underlying idea is similar.

**Label-free performance estimation** Similarly, our approach is also related to the general fields of performance estimation without labels Garg et al. (2020); Guillory et al. (2021); Chen* et al. (2021) and active testing Kossen et al. (2021). Part of the problem is similar in that the goal is to estimate performance; however, the similarity ends there, as these methods generally assume access to the model $f$ for which performance is estimated, as well as access to the features of the dataset Garg et al. (2020). Our approach involves forecasting performance not only for known models but also for unknown models. While our approach does not explicitly differentiate strategies, it is true that

we have access to additional information. Therefore, extensions that leverage existing techniques in this area could strengthen our method.

This forecasting problem can seem similar to the problem of uncertainty quantification Hendrycks & Gimpel (2017); Liu et al. (2020), but we are targeting average performance of unknown models, not the probability of error of a given model at a given input $P(f(x) = y|x)$.

**Offline reinforcement learning** Lastly, we discuss the connection to the offline reinforcement learning (ORL) setting, where the agent must learn a policy from a fixed dataset of rewards, actions, and states. This subset of RL is particularly challenging, as the agent cannot explore the entire MDP and can only rely on the dataset to infer the underlying dynamics and handle distribution shifts (Ross et al., 2011; Levine et al., 2020; Hejna et al., 2023). Policy gradient methods can be adapted to the offline setting using variants of importance sampling, but they are generally prone to high variance and require large amounts of data to be effective (Levine et al., 2020). For this reason, Q-learning and value function methods, where the task is to predict the future costs of actions, have emerged as the preferred approaches for ORL (Levine et al., 2020; Kalashnikov et al.; Hejna et al., 2023; Kostrikov et al., 2022). Lagoudakis & Parr (2003) presents a classical method that uses a linear approximation of the Q-function, while (Kalashnikov et al.) employs convolution-based Q-function architectures for vision tasks.Others have leveraged advancements in sequential learning, applying transformer-based architectures to predict rewards(Janner et al., 2021) or Q-functions(Chebotar et al., 2023). Some methods integrates epistemic uncertainty on Q-function to account for the distribution shift of unseen actions (Kumar et al., 2020; O'Donoghue et al., 2017; Luis et al., 2023).

If we view the states as time and the model in use, and actions as either retraining or maintaining the current model, we can frame this problem as an offline reinforcement learning (RL) problem. The problem would also feature a deterministic transition matrix and a highly structured reward which unusual in RL. However, most existing approaches focus on scaling to very large state spaces, employing large models, and assuming access to abundant data, making them unsuitable for our context. A key requirement for our approach is that it must be highly efficient to train. If the resources required for making a retraining decision are comparable to those for retraining the model itself, the approach becomes impractical.

## 8.2 PROOF OF PROPOSITION 3.1

We provide the proof for our result from Proposition 3.1, which states the following.

Given that $L \geq |pe_{i,t} - pe_{i+1,t}| \ \forall t \in [T]$, a horizon of $T \in \mathbb{N}$, and a relative cost of retrain $\alpha$, the number of retrains of the solution to Equation 5 $r^* \triangleq \|\boldsymbol{\theta}^*\|_1$ satisfies:

$$r^* \leq T - \sqrt{\frac{\alpha}{L}} \tag{22}$$

We start by defining a function that takes the model index $i$ and the timesteps $t$ as arguments, and outputs the performance $pe(i, t) = pe_{i,t}$, and rewrite the objective:

$$C_\alpha(\boldsymbol{\theta}) = \alpha \|\boldsymbol{\theta}\|_1 + \sum_{t=1}^{T} pe\left(r_{\boldsymbol{\theta}}(t), t\right), \tag{23}$$

$$\boldsymbol{\theta}^* = \underset{\boldsymbol{\theta} \in \{0,1\}^T}{\arg\min} \, C_\alpha(\boldsymbol{\theta}), \tag{24}$$

where we still have that $r_{\boldsymbol{\theta}}(t)$ returns the most recent index of retraining at $t$.

**Subproblem with a fixed number of retrains** We can break down this optimization problem into subproblems, where we solve for the optimal retraining schedule for a given fixed number of retrains $r$. We define such a subproblem as follows:

$$C_r(\boldsymbol{\theta}) = \alpha r + \sum_{t=1}^{T} pe\left(r_{\boldsymbol{\theta}}(t), t\right), \tag{25}$$

$$\boldsymbol{\theta}_r^* = \underset{\boldsymbol{\theta} \in \{0,1\}^T \, s.t. \, \|\boldsymbol{\theta}\|=r}{\arg\min} \, C_r(\boldsymbol{\theta}). \tag{26}$$

Since we know that we will have $r$ retrains, we can rewrite this subproblem by encoding the retraining decisions as $r$ timesteps of retrain $t_1 < \cdots < t_r$. We use a simple index mapping function $I : [T]^r \to \{0,1\}^T$:

$$I(\{t_1, \ldots, t_r\}) = \boldsymbol{\theta} \ s.t. \begin{cases} \boldsymbol{\theta}_t = 1 \text{ if } t \in \{t_1, \ldots, t_r\} \\ \boldsymbol{\theta}_t = 0 \text{ o.w.} \end{cases} \tag{27}$$

We can remove the constant $\alpha r$ from the objective as it does not depend on the parameters anymore. The solution of Eqn 26 is given by:

$$\boldsymbol{\theta}_r^* = \underset{\boldsymbol{\theta} \in \{0,1\}^T \ s.t. \|\boldsymbol{\theta}\|=r}{\arg\min} \alpha r + \sum_{t=1}^T pe\left(r_{\boldsymbol{\theta}}(t), t\right) \tag{28}$$

$$= \underset{\boldsymbol{\theta} \in \{0,1\}^T \ s.t. \|\boldsymbol{\theta}\|=r}{\arg\min} \sum_{t=1}^T pe\left(r_{\boldsymbol{\theta}}(t), t\right) \text{ since the } \alpha r \text{ is fixed} \tag{29}$$

$$= I\left(\underset{t_1 < \cdots < t_r \in [T]^r}{\arg\min} \sum_{s=1}^{t_1} pe(0, s) + \sum_{i=1}^{r-1}\left(\sum_{s=t_i}^{t_{i+1}} pe(t_i, s)\right) + \sum_{s=t_r}^T pe(t_r, s)\right) \tag{30}$$

$$\boldsymbol{\theta}_r^* = I\left(\underset{t_1 < \cdots < t_r \in [T]^r}{\arg\min} M_r(\{t_1, \ldots, t_r\})\right) \tag{31}$$

$$\text{where } M_r(\{t_1, \ldots, t_r\}) \triangleq \sum_{s=1}^{t_1} pe(0, s) + \sum_{i=1}^{r-1}\left(\sum_{s=t_i}^{t_{i+1}} pe(t_i, s)\right) + \sum_{s=t_r}^T pe(t_r, s) \tag{32}$$

We therefore can focus on the new objective $M_r(\{t_1, \ldots, t_r\})$ as minimizing this objective is equivalent to finding $\boldsymbol{\theta}_r^*$.

$$\{t_1, \ldots, t_r\}^* = \underset{t_1 < \cdots < t_r \in [T]^r}{\arg\min} M_r(\{t_1, \ldots, t_r\}) \tag{33}$$

$$M_r^* \triangleq M_r(\{t_1, \ldots, t_r\}^*) \tag{34}$$

$$\boldsymbol{\theta}_r^* = I\left(\{t_1, \ldots, t_r\}^*\right) \tag{35}$$

**Lemma 8.1.** *Given a discrete function $pe : [T] \times [T] \to \mathbb{R}$ with bounded $L \geq |pe(i,t) - pe(i+1,t)|$, a timestep horizon $T \in \mathbb{N}$, and a number of retrains $r \in \{1, T-1\}$, we can show that:*

$$L(T-r)^2 \geq M_r^* - M_{r+1}^* \tag{36}$$

*That is, the relative improvement of performance cost that you can gain by increasing the number of retrainings from $r$ to $r+1$ is upper bounded by $L(T-r)^2$.*

This allows us to preemptively determine the maximum number of retains $r$ we have to consider for solving our initial problem, as we know the cost of adding one more retrain ($\alpha$). Therefore, once $L(T-r)^2$ is smaller than $\alpha$, the optimal solution cannot have higher than $r$ retrains. That is,

$$L(T - r^*)^2 < \alpha \implies r^* < T - \sqrt{\frac{\alpha}{L}} \quad \square \tag{37}$$

This concludes our proof for Proposition 8.2. We provide the proof for Lemma 8.1 in the following section.

**Proof Lemma 8.1** To prove this lemma, we decompose the $M^*_{r+1}$ quantity into the $M_r$ value we would obtain with the first $r$ timesteps of the solution $\{t_1, \ldots t_{r+1}\}^*$ and some value:

$$M^*_{r+1} = \sum_{s=1}^{t^*_1} pe(0,s) + \sum_{i=1}^{r-1} \left( \sum_{s=t^*_i}^{t^*_{i+1}} pe(t^*_i, s) \right) + \sum_{s=t^*_r}^{t^*_{r+1}} pe(t^*_r, s) + \sum_{s=t^*_{r+1}}^{T} p(t^*_{r+1}, s) \tag{38}$$

$$= \sum_{s=1}^{t^*_1} pe(0,s) + \sum_{i=1}^{r-1} \left( \sum_{s=t^*_i}^{t^*_{i+1}} pe(t^*_i, s) \right) + \sum_{s=t^*_r}^{t^*_{r+1}} pe(t^*_r, s) + \sum_{s=t^*_{r+1}}^{T} pe(t^*_r, s) \tag{39}$$

$$- \sum_{s=t^*_{r+1}}^{T} pe(t^*_r, s) + \sum_{s=t^*_{r+1}}^{T} pe(t^*_{r+1}, s) \text{ adding } 0 \tag{40}$$

$$= \sum_{s=1}^{t^*_1} pe(0,s) + \sum_{i=1}^{r-1} \left( \sum_{s=t^*_i}^{t^*_{i+1}} pe(t^*_i, s) \right) + \sum_{s=t^*_r}^{T} pe(t^*_r, s) \tag{41}$$

$$- \sum_{s=t^*_{r+1}}^{T} pe(t^*_r, s) + \sum_{s=t^*_{r+1}}^{T} pe(t^*_{r+1}, s) \text{ combining the sum over } pe(t^*_i, s) \tag{42}$$

$$M^*_{r+1} = M_r(\{t_1, \ldots t_{r+1}\}^* \smallsetminus t^*_{r+1}) - \sum_{s=t^*_{r+1}}^{T} pe(t^*_r, s) + \sum_{s=t^*_{r+1}}^{T} pe(t^*_{r+1}, s) \tag{43}$$

$$= M_r(\{t_1, \ldots t_{r+1}\}^* \smallsetminus t^*_{r+1}) - \sum_{s=t^*_{r+1}}^{T} (p(t^*_r, s) - pe(t^*_{r+1}, s)). \tag{44}$$

By definition, we know that;

$$M_r(\{t_1, \ldots t_{r+1}\}^* \smallsetminus t^*_{r+1}) \geq M^*_r. \tag{45}$$

That is, the $M$ value that we obtain by removing the last timestamp using the solution for the $r + 1$ problem. Using that inequality in our previous result, we obtain the final result;

$$M^*_{r+1} \geq M^*_r - \sum_{s=t^*_{r+1}}^{T} (pe(t^*_r, s) - pe(t^*_{r+1}, s)) \tag{46}$$

$$\geq M^*_r - \sum_{s=t^*_{r+1}}^{T} L(t^*_{r+1} - t^*_r) \tag{47}$$

$$\geq M^*_r - (T - t^*_{r+1}) L(t^*_{r+1} - t^*_r) \tag{48}$$

$$M^*_{r+1} \geq M^*_r - L(T - r)^2. \tag{49}$$

$\square$

### 8.2.1 PROPOSITION 3.1 IN PRACTICE

In this section, we illustrate how to use the result from Proposition 3.1 in practice. To restate, proposition states the following;

Given that $L \geq |pe_{i,t} - pe_{i+1,t}| \; \forall t \in [T]$, a horizon of $T \in \mathbb{N}$, and a relative cost of retrain $\alpha$, the number of retrains of the solution to Equation 5 $r^* \triangleq \|\boldsymbol{\theta}^*\|_1$ satisfies:

$$r^* \leq T - \sqrt{\frac{\alpha}{L}}. \tag{50}$$

We present the $\alpha$ values that guarantee various numbers of optimal retrains $r^* = 0, 1, 2$ in our experiment. Since we can't provide a true upper bound for the $L$ value, we approximate it using the empirical maximum value that we observe in a specific dataset for $|pe_{i,t} - pe_{i+1,t}|$. In Figure 4, we can see that the $\alpha$ at which we know for certain that we don't need to retrain is not too far off the operational region of the problem. The oracle decides to not retrain around $\alpha = 0.5$, and the bound from our result guarantees that we don't have to retrain if the selected $\alpha$ is larger than $0.96$.

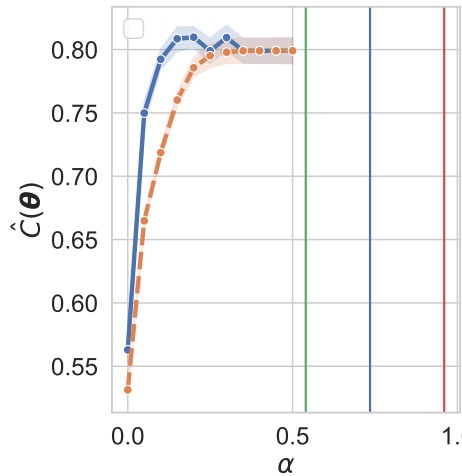 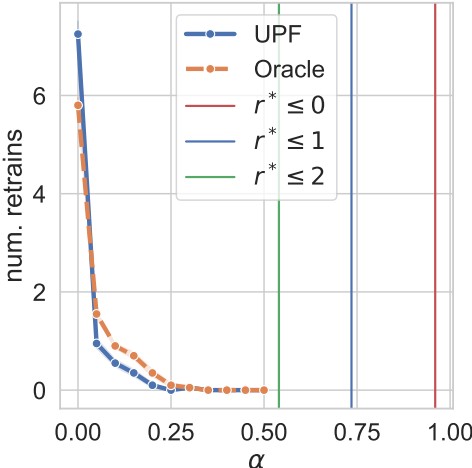

Figure 4: Results on the Gauss dataset, with the $\alpha$ values from Proposition 8.1 providing different upper bounds on the optimal number of retrain $r^*$. **Left)** Cost $\hat{C}_\alpha(\boldsymbol{\theta})$ vs $\alpha$. **Right)** Number of retrains vs $\alpha$.

### 8.3 BOUNDING $L$

In this section, we provide more details on the known results from the literature that can be connected to the bound $L$.

**Approximating $L$ from known upper bounds** For some simple models, explicit bounds on the expected performance as a function of the number of samples $N$ have been derived. We can use those upper bounds to approximate $L$ under no distribution shift, where the dataset size is steadily increasing by a known number of samples $|\mathcal{D}|$.

**Theorem 8.2** (Standard generalization in the Gaussian model (from (Schmidt et al., 2018) )). *Let* $(x_1, y_1), \ldots, (x_{(i+1)|\mathcal{D}|}, y_{(i+1)|\mathcal{D}|}) \in \mathbb{R}^d \times \{\pm 1\}$ *be drawn i.i.d. from a* $(\theta^*, \sigma)$*-Gaussian model with* $\|\theta^*\|_2 = \sqrt{d}$. *Let* $\hat{w} \in \mathbb{R}^d$ *be the unit vector in the direction of* $\overline{z} = \frac{1}{(i+1)|\mathcal{D}|} \sum_{i=1}^{(i+1)|\mathcal{D}|} y_i x_i$, *i.e.,* $\hat{w} = \overline{z}/\|\overline{z}\|_2$. *Then with probability at least* $1 - 2\exp\left(-\frac{d}{8(\sigma^2+1)}\right)$, *the linear classifier* $f_{\hat{w}}$ *has classification error at most;*

$$pe_{i,t} \le \exp\left(-\frac{(2\sqrt{(i+1)|\mathcal{D}|}-1)^2 d}{2(2\sqrt{(i+1)|\mathcal{D}|}+4\sigma)^2\sigma^2}\right). \tag{51}$$

For the proof please refer to (Schmidt et al., 2018). An $L$ bound value can therefore be loosely approximated to match the gap of the upper bound;

$$|pe_{i,t} - pe_{i+1,t}| < L \approx \exp\left(-\frac{(2\sqrt{(i+1)|\mathcal{D}|}-1)^2 d}{2(2\sqrt{(i+1)|\mathcal{D}|}+4\sigma)^2\sigma^2}\right) - \exp\left(-\frac{(2\sqrt{(i+2)|\mathcal{D}|}-1)^2 d}{2(2\sqrt{(i+2)|\mathcal{D}|}+4\sigma)^2\sigma^2}\right). \tag{52}$$

**Beyond IID data.** In real-world applications, data often exhibits temporal or spatial dependencies, making the non-distribution shift i.i.d. assumption unrealistic. For non-i.i.d. processes, stability analysis (Mohri & Rostamizadeh, 2007; 2010) or bounds based on Rademacher complexity (Mohri & Rostamizadeh, 2008) can be used to analyze generalization performance and thus to derive retraining schedules in more complex scenarios.

In the context, of the proposed retraining framework, bounds like this theoretically allow us to make precise statements about the benefit of retraining $L$ to derive optimal retraining schedules. In

practice, deriving a retraining schedule from these bounds would provide a loose and non-sufficient estimate. Thus, we introduce a data-driven algorithm to estimate optimal retraining schedules in our work.

**Empirical knowledge on the scaling law of $L$ on $N$ for LLMs** Kaplan et al. (2020) derive scaling laws for large language models (LLMs) concerning the dependency of the final cross-entropy loss depending on model size, dataset size and compute budget used for training. They find a power-law for all of the aforementioned parameters. For example, they find that the loss $\mathcal{L}$ of the neural network scales with respect to the dataset size $N$ as $\mathcal{L} = \left(N/5.4 \cdot 10^{13}\right)^{-0.095}$. This empirical relationship provides valuable insights for determining optimal retraining schedules. By quantifying how loss decreases with increasing dataset size, it enables researchers to estimate the expected performance improvements from expanded datasets $L$ and to make informed decisions about when retraining would yield substantial benefits.

## 8.4 DATASET

Dataset statistics can be viewed in Table 4.

Table 4: Dataset description. $w$ denotes the number of timestep of the offline phase, $T$ denotes the number of timestep of the online phase. The Model describes the architecture used for each $f_t$.

| Dataset | Model | $\alpha_{max}$ | w | $|\mathcal{M}_{<0}|$ | T | Dataset size ($|D|$) | Num. features | Task | Total N |
|---|---|---|---|---|---|---|---|---|---|
| Gauss | XGBoost | 0.5 | 7 | 21 | 8 | 5000 | 2 | Binary | - (Synthetic) |
| circles | XGBoost | 0.25 | 7 | 21 | 8 | 5000 | 2 | Binary | - (Synthetic) |
| electricity | XGBoost | 1 | 7 | 21 | 8 | 2000 | 6 | Binary | 4,5312 |
| yelpCHI | XGBoost | 0.1 | 7 | 21 | 8 | 4000 | 25 | Binary | 67,395 |
| epicgames | XGBoost | 0.1 | 7 | 21 | 8 | 1000 | 400 | Binary | 17,584 |
| airplanes | XGBoost | 0.7 | 7 | 21 | 8 | 3000 | 7 | Binary | .. |
| iWild | Vision Model (see 8.4.1) | 1 | 7 | 21 | 8 | 40,605 | 224x224+1 | 100 | 539,383 |

In this section, we provide a more detailed overview of each retraining datasets. Except for the iWild experiment, each individual dataset $\mathcal{D}_t$ is constructed with distinct samples, with no overlap between $\mathcal{D}_t$ and $\mathcal{D}_{t-1}$. For the electricity, airplanes, yelpCHI, and epicgames datasets, the partitions are determined based on the timestamp of each sample (i.e., the datasets are divided in temporal sequence).

- **electricity** (Harries et al.) is a binary classification where the task is to predict the rise or fall of electricity prices in New South Wales, Australia. The distribution evolve due to change in consumption patterns.

- **airplanes** (Gomes et al., 2017) is also a binary task where the task is to predict if a flight will be delayed. We follow Mahadevan & Mathioudakis (2024) and use the Sklearn Multiflow library version (Montiel et al., 2018) of the airplane dataset.

- **yelpCHI** (Dou et al., 2020) is a spam dataset. The dataset contains users, hotels and restaurants. An interaction occurs when a user submits a review for one of these hotels or restaurants. Reviews are categorized as either filtered (indicating spam) or recommended (indicating legitimate content).

- **epicgames** (Ozmen et al., 2024) includes critiques from authors on games released on the epicgames platform. Interaction features are created by vectorizing the critiques using TF-IDF and incorporating the author's overall rating. The interaction label indicates whether the critique was chosen as a top critique.

- **Gauss** is a 2 dimensional synthetic dataset. The input features as generated as $X_t \sim \mathcal{N}(\mu_1(t), \mu_2(t), \sigma \mathbf{1})$ where $\mu_1(t) = \frac{(t+1)}{100}$, $\mu_2(t) = 0.5 - \frac{(t+1)}{100}$, $\sigma = 0.1$. The label is generated using a fixed rule $y = \mathbb{1}[4 * \mathbf{r}_1 - 0.5) * *2 > \mathbf{r}_2]$.

- **circles** is a 2 dimensional synthetic dataset. The input features as uniformly generated as $X_t \sim U[0, 1]$ The label is generated using a moving rule $y_t = \mathbb{1}[(\mathbf{r}_1 - (0.2 + 0.02t))^2 + (\mathbf{r}_2 - (0.2 + 0.02t))^2 \leq 0.5 \in]$.

- **iWild** (Beery et al., 2020) is a multiclass dataset featuring images of animals captured in the wild at various locations. Originally used as a domain transfer benchmark, we adapted it into a standard classification dataset by including the location ID as a feature for the model. To obtain a long enough sequence of datasets $\mathcal{D}_0, \mathcal{D}_1, \ldots$, we create the individual datasets $\mathcal{D}_i$ using overlapping windows on the timeframe, i.e., half of the most recent images in $\mathcal{D}_i$ are contained in $\mathcal{D}_{i+1}$. We avoid data leakage by ensuring that the train/val/test splits are maintained.

### 8.4.1 BASE MODEL OF THE IWILD DATASET

To motivate our cost considerations, we present an experiment where the base model architecture is not fixed and is searched for across a list of potential model architectures. This could happen in practice for important applications; nothing forces a practitioner to use the same base model $f$ at each timestep.

Our architecture involves using a pretrained vision model, with a new output layer added to match the correct number of classes for our task, which is then fine-tuned for up to 20 epochs. The fine-tuning process uses the Adam optimizer with a fixed learning rate of $10^{-4}$ and a weight decay parameter of $10^{-5}$. Training was conducted using 4 H100 GPUs for 2 days.

At each timestep $f_t$, we perform a random search over the pretrained vision models made available from `timm`, which includes 188 vision models of varying configuration and base architecture. We include the list in Appendix 8.13. We also include in our search the option to early stop or not, using the validation set. The model used for $f_t$ is the one that obtains the best validation accuracy.

### 8.5 PERFORMANCE FORECASTER

In this section, we provide additional details on the proposed algorithm to forecast the performance.

To restate, instead of learning the $\alpha(\mathbf{r}_{i,j}), \beta(\mathbf{r}_{i,j})$ parameters, we learn the mean and variance parameters;

$$\mu(\mathbf{r}_{i,j}) \tag{53}$$
$$\sigma(\mathbf{r}_{i,j}). \tag{54}$$

And convert the learned parameters to the parameters of a beta distribution using the following relation (with appropriate clipping if needed):

$$\alpha = \mu\left(\frac{\mu(1-\mu)}{\sigma^2} - 1\right) \tag{55}$$

$$\beta = (1 - \mu)\left(\frac{\mu(1-\mu)}{\sigma^2} - 1\right) \tag{56}$$

**Inputs $\mathbf{r}_{i,j}$** As stated, the input of our performance forecaster model contains the model index $i$, the timesteps $j$, the time since retrain $j-i$ and summary statistics of the distribution shift $z_{shift}$. $z_{shift}$ is constructed by taking the average feature shift between the features of the most recently available subsequent datasets $\mathcal{D}_t$ and $\mathcal{D}_{t-1}$ (where $t$ denotes the time step of the most recent available dataset). We compute the mean features of each dimension for a given dataset; $\bar{\mathbf{x}} = \frac{1}{|\mathcal{D}_t|} \sum_{i=1}^{|\mathcal{D}_t|} x_i$ and compute the $\ell_1$ distance between the mean feature vector of the two subsequent datasets;

$$z_{shift} = \|\bar{\mathbf{x}}_t - \bar{\mathbf{x}}_{t-1}\|_1 \tag{57}$$

The input features are thus given by concatenating $\mathbf{r}_{i,j} = [i, j, j - i, z_{shift}]$.

Since our methodology involves forecasting the performance of future models and on future datasets to be used by our decision algorithm, we assess the regression performance of our forecasting models and analyze how it impacts the overall performance of our UPF algorithm.

To do so, we construct two versions of our forecaster module $\mu_\phi(\mathbf{r}_{i,j})$ that are designed to be less performant than our proposed method.

- **UPF overfit:** A baseline designed to overfit the training data. We use a Gaussian Process-based $\mu_\phi(\mathbf{r}_{i,j})$ with no white noise kernel, using a single dot product kernel from `scikit-learn`.
- **UPF overfit+noise:** This variant further decreases performance by using the same overfitting model and adding random noise to the target values.

We report two metrics, the average mean absolute error of our prediction $\mu$ and the average bias of our prediction $\mu_\phi(\mathbf{r}_{i,j}) - a_{i,j}$ on the test set. We start by reporting the retraining performance of each baseline w.r.t. our base retraining metric, the AUC of cost values evaluated at different $\alpha$ in Table 5. As expected, the best performing method is the method with our proposed UPF baseline which is expected to reach the best MAE error on it's performance prediction, on all datasets.

Table 5: AUC of the combined performance/retraining cost metric $\hat{C}_\alpha(\boldsymbol{\theta})$, computed over a range of $\alpha$ values, for all datasets. The bolded entries represent the best, and the underlined entries indicate the second best. The $*$ denotes statistical significance with respect to the next best baseline, evaluated using a Wilcoxon test at the $5\%$ significance level.

|  | Gauss | circles | epicgames | electricity | yelp | airplanes |
|---|---|---|---|---|---|---|
| UPF overfit+noise | 0.3845 | 0.0722 | 0.3253 | 2.6389 | 0.1194 | 2.3767 |
| UPF overfit | 0.3849 | 0.0663 | 0.3224 | 2.6001 | 0.1194 | 2.3352 |
| UPF | **0.3836**\* | **0.0662**\* | **0.3203**\* | **2.5910**\* | **0.1175**\* | **2.3094**\* |

We then visualize the effect of the performance forecasting precision (measured with MAE and bias) on the decision algorithm's performance (measured by $\hat{C}_\alpha(\boldsymbol{\theta})$) in the following figures.

Overall, we observe that the impact of poor performance depends on the difficulty of the underlying dataset.

For the airplane dataset, which is of standard difficulty, we can observe a gradual impact of the degradation in forecasting performance on the overall retraining metric in Figure 5. The best MAE leads to the best cost metric $\hat{C}_\alpha(\boldsymbol{\theta})$, and the performance gradually decreases as the MAE and bias worsen.

The Epicgame dataset 6, which is more challenging due to its less regular performance trends, shows a different behavior. Here, the overall forecasting performance is worse (the best achievable MAE is higher), and we observe a less regular pattern where poorer MAE does not always result in a proportional increase in cost, as shown in terms of scale. Similarly, when turning to the synthetic datasets, the circle dataset, which is constructed with concept drift (changing $p(Y|X)$), is more challenging than the Gauss dataset, which only exhibits feature drift (where $p(X)$ changes, but $p(Y|X)$ remains constant). This impacts the effect of poor forecasting performance. In Figure 7, for the circle dataset, we observe that a small decrease in MAE paired with stronger bias can have a more sudden and drastic effect on the decision policy. Conversely, in the Gauss dataset (Figure 8), the effect of poorer forecasting performance is less pronounced.

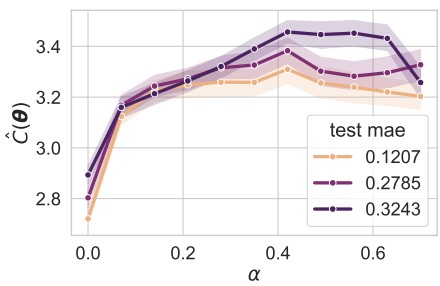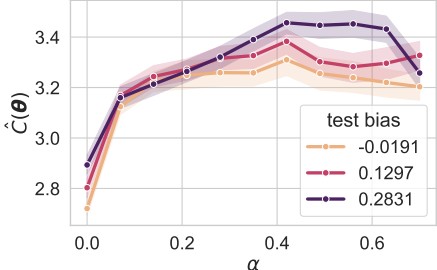

Figure 5: Airplanes. Cost $\hat{C}_\alpha(\boldsymbol{\theta})$ vs $\alpha$ with the forecasting performance metrics (mae and bias).

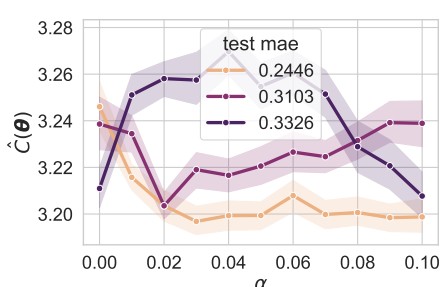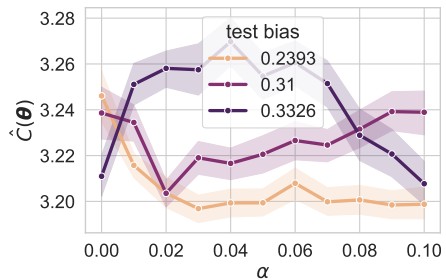

Figure 6: Epicgames. Cost $\hat{C}_\alpha(\boldsymbol{\theta})$ vs $\alpha$ with the forecasting performance metrics (mae and bias).

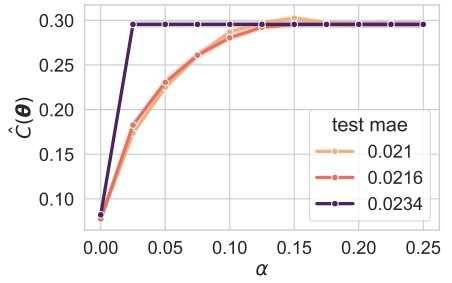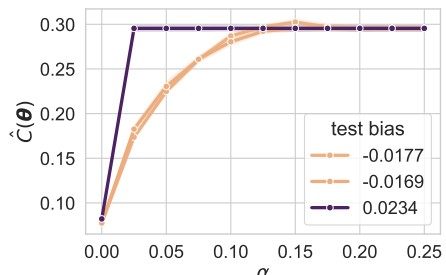

Figure 7: Circles. Cost $\hat{C}_\alpha(\boldsymbol{\theta})$ vs $\alpha$ with the forecasting performance metrics (mae and bias).

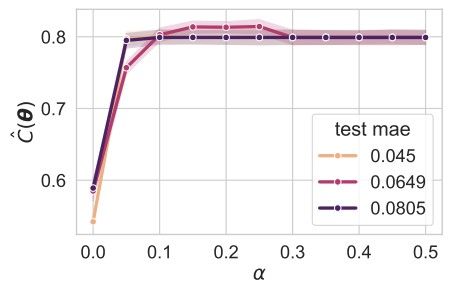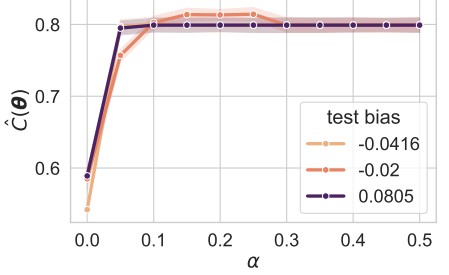

Figure 8: Gauss. Cost $\hat{C}_\alpha(\boldsymbol{\theta})$ vs $\alpha$ with the forecasting performance metrics (mae and bias).

## 8.6 EXTENSION TO NON-BOUNDED METRICS

In this section, we show how we can extend our methodology to model non-bounded metrics often used in regression tasks, such as the root mean square error (RMSE) or mean absolute error (MAE).

To do so, we replace the use of a Beta distribution to a log Normal distribution to model our performance metric r.v. $A_{i,j}$.

A log normal distribution is parameterized with location $m$ and scale parameter $v$. We can learn the mean and variance parameters using the same Gaussian approximation;

$$\text{LogNorm}(m(\mathbf{r}_{i,j}), v(\mathbf{r}_{i,j})) \approx \mathcal{N}(\mu(\mathbf{r}_{i,j}), \sigma(\mathbf{r}_{i,j})), \tag{58}$$

and recover the location and scale parameters using the relation;

$$v = \sqrt{\ln(1 + \frac{\mu}{\sigma^2})} \tag{59}$$

$$m = \ln(v) - \frac{v^2}{2}. \tag{60}$$

### 8.6.1 IMPACT OF THE NORMAL APPROXIMATION

In our method, we approximate the Beta distribution with a Normal distribution to ease the learning process;

$$Beta(\alpha(\mathbf{r}_{i,j}), \beta(\mathbf{r}_{i,j})) \approx \mathcal{N}(\mu(\mathbf{r}_{i,j}), \sigma(\mathbf{r}_{i,j})). \tag{61}$$

We verify here that this approximation doesn't have too big an effect on the end performance. We compare the **UPF** method, which uses $A_{i,j} \sim \text{Beta}(\alpha(\mathbf{r}_{i,j}), \beta(\mathbf{r}_{i,j}))$, with a **UPF (Gaussian)**, which doesn't use the Beta distribution and instead uses a Gaussian with learned parameters to model the performance metric: $A_{i,j} \sim \mathcal{N}(\mu(\mathbf{r}_{i,j}), \sigma(\mathbf{r}_{i,j}))$. In Figures 9, 10, 11 and 12, we can see that this does not have too big an effect on the overall behavior and performance.

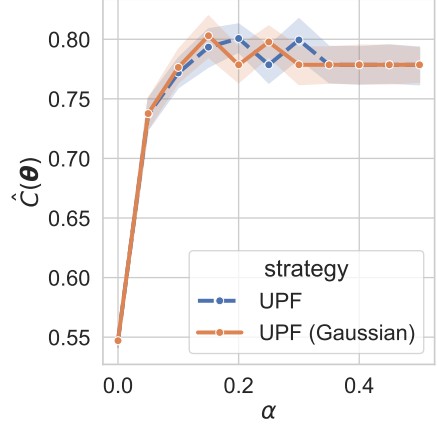

Figure 9: Gauss

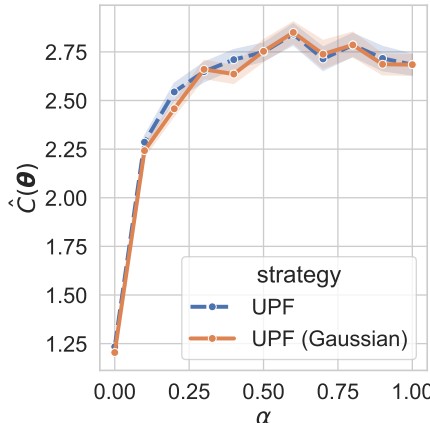

Figure 10: Electricity

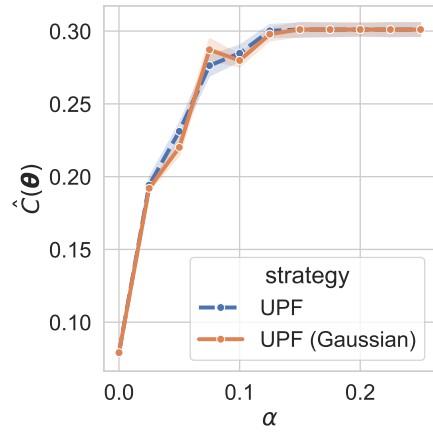

Figure 11: Circles

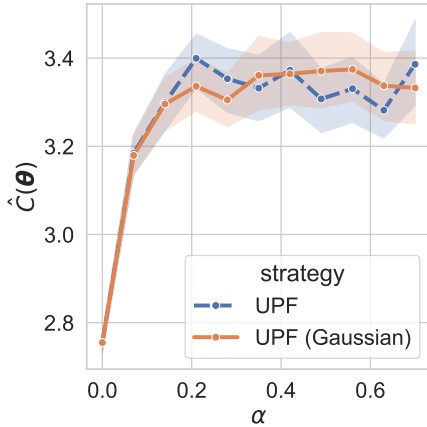

Figure 12: Airplanes

## 8.7 TRAINING COMPLEXITY

In this section, we compare the training complexity of each baseline. We report the average time required for the offline training process, online inference and discuss runtime complexity.

The CARA baseline comprises two computationally intensive components. First, it constructs the $C$ matrix, representing its performance estimation. This algorithm involves inferring, with a modified model, each point of the new dataset and reweighting each, which scales with $\mathcal{O}(|\mathcal{D}_{\text{new}}|)$. This needs to be done in both offline and online phases. Then, in the offline phase, it performs an annealing search over parameters to find the best value that minimizes this cost approximation, taking into account the retraining cost associated with each decision. In Table 6, we can see that this result in the highest runtime for both online and offline phases.

Table 6: Average runtime of the baselines on the circles dataset.

|                     | CARA cum. | CARA   | CARA per. | UPF    | ADWIN  | FHDDM  | KSWIN  |
|---------------------|-----------|--------|-----------|--------|--------|--------|--------|
| Offline ms          | 8.4871    | 8.6608 | 7.8461    | 0.0947 | 0.0274 | 0.0122 | 0.3392 |
| Online (one step)ms | 1.5604    | 1.5046 | 1.5940    | 0.0247 | 0.0351 | 0.0103 | 0.3438 |

In comparison, our approach consists of fitting a linear model on a small dataset. The shift distribution features must be obtained, but they involve comparing two histograms, scaling as $\mathcal{O}(w^2|\mathcal{D}_t|)$ rather than exponentially with $|\mathcal{D}_t|$.

The distribution shift baselines do not have an offline phase, as they monitor shifts in the underlying distribution continuously. Their runtime complexity is therefore very low, at $\mathcal{O}(|\mathcal{D}_t|)$, as reflected in Table 6

## 8.8 ADDITIONAL RESULTS

In this section, we include additional figures to visualize our results in Figures 13, 14, 15, 16, 17, 18, and Figures 19. Overall, the results are generally consistent and exhibit a similar trend. The EpicGames dataset, however, is more challenging and presents greater difficulties for all baselines. In particular, UPF performs worse than other baselines at low values of the retraining cost ratio $\alpha$. For those operating points, UPF does reach the correct retraining frequency; however, it is unable to pinpoint the optimal moments to retrain, resulting in worse performance than baselines that retrain more frequently, as shown in the right panel of Figure 19.

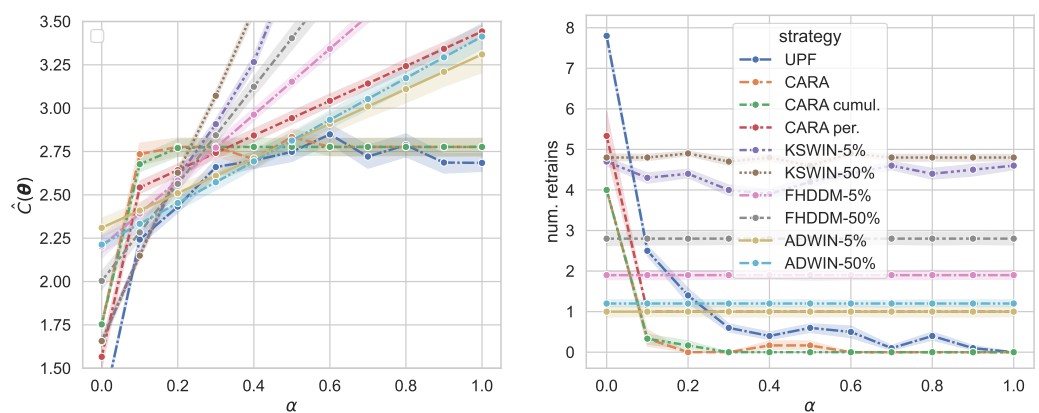

Figure 13: Result on the electricity dataset. **left)** Cost $\hat{C}_\alpha(\boldsymbol{\theta})$ vs $\alpha$. **right)** Number of retrains vs $\alpha$.

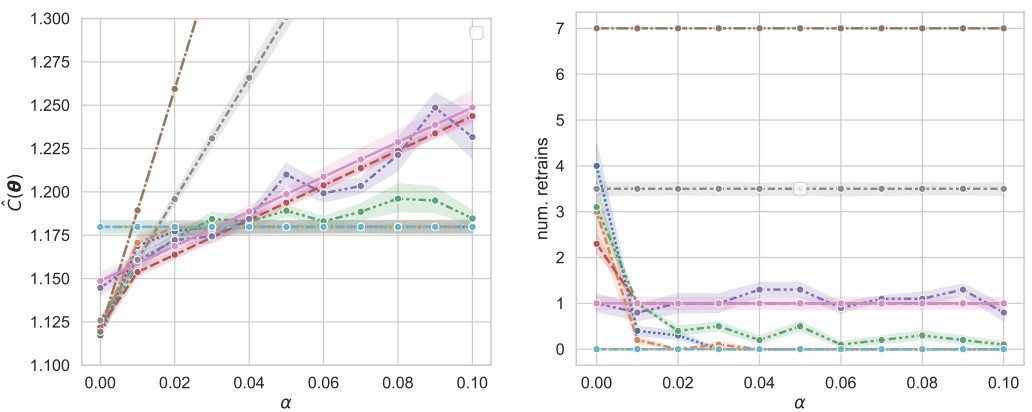

Figure 14: Result on the yelp dataset. **left)** Cost $\hat{C}_\alpha(\boldsymbol{\theta})$ vs $\alpha$. **right)** Number of retrains vs $\alpha$.

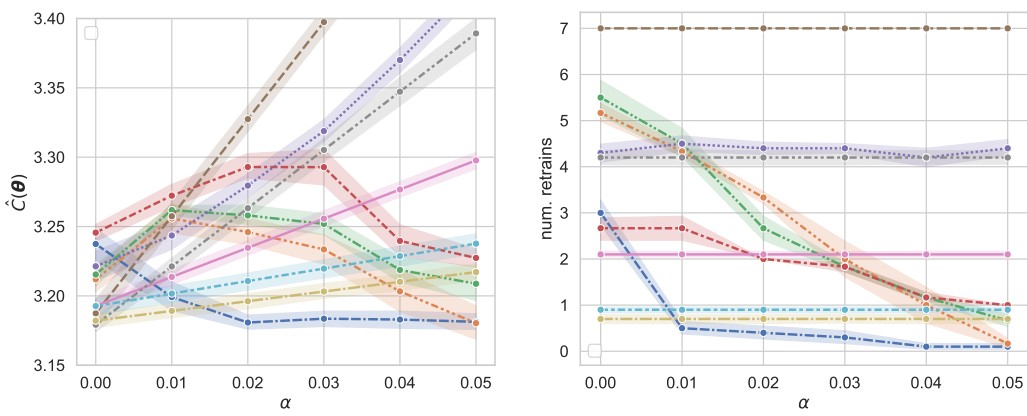

Figure 15: Result on the epicgames dataset. **left)** Cost $\hat{C}_\alpha(\boldsymbol{\theta})$ vs $\alpha$. **right)** Number of retrains vs $\alpha$.

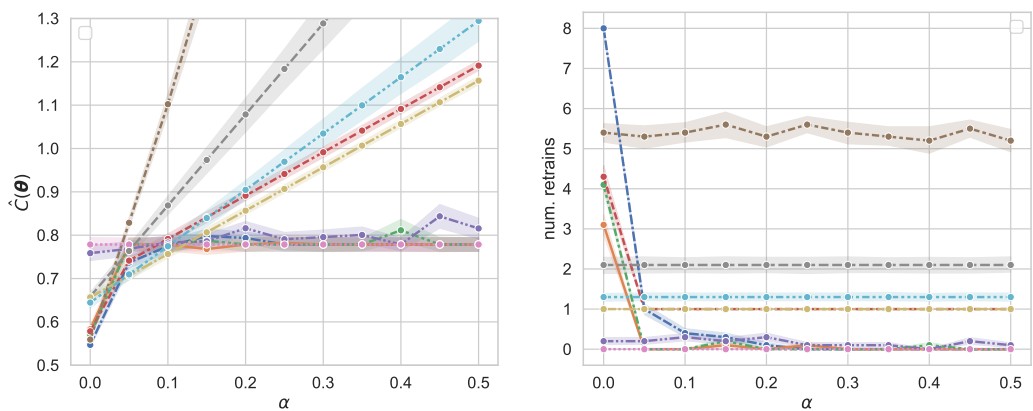

Figure 16: Result on the Gauss dataset. **left)** Cost $\hat{C}_\alpha(\boldsymbol{\theta})$ vs $\alpha$. **right)** Number of retrains vs $\alpha$.

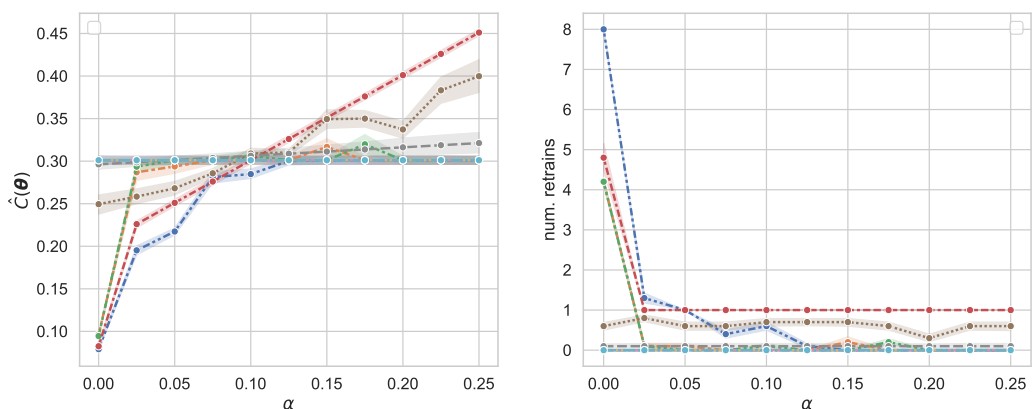

Figure 17: Result on the circles dataset. **left)** Cost $\hat{C}_\alpha(\boldsymbol{\theta})$ vs $\alpha$. **right)** Number of retrains vs $\alpha$.

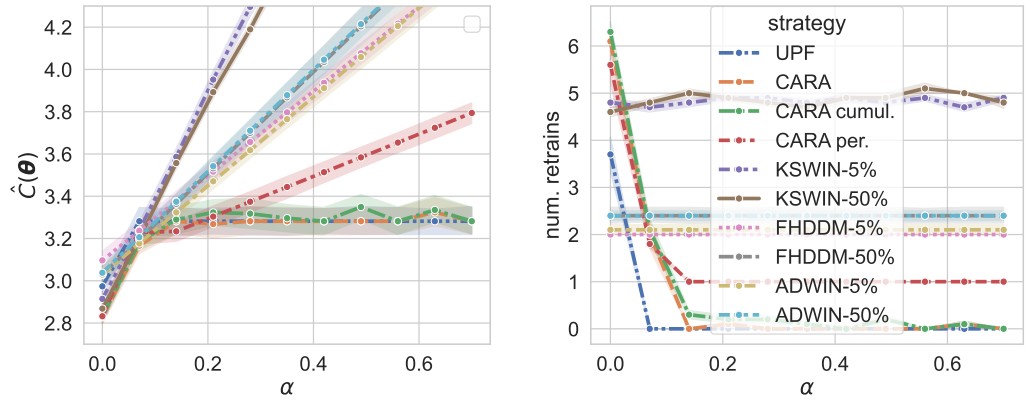

Figure 18: Result on the airplanes dataset. **left)** Cost $\hat{C}_\alpha(\boldsymbol{\theta})$ vs $\alpha$. **right)** Number of retrains vs $\alpha$.

We additionally include results with the oracle baselines in Figures 19. We can see that the UPF baseline is reasonably close to the optimal algorithm in two of the datasets (circles and electricity),

but struggles for the more challenging dataset, epicgames. Looking at the number of retrains, we can see that UPF more closely follows the retraining frequency of the oracle for all datasets.

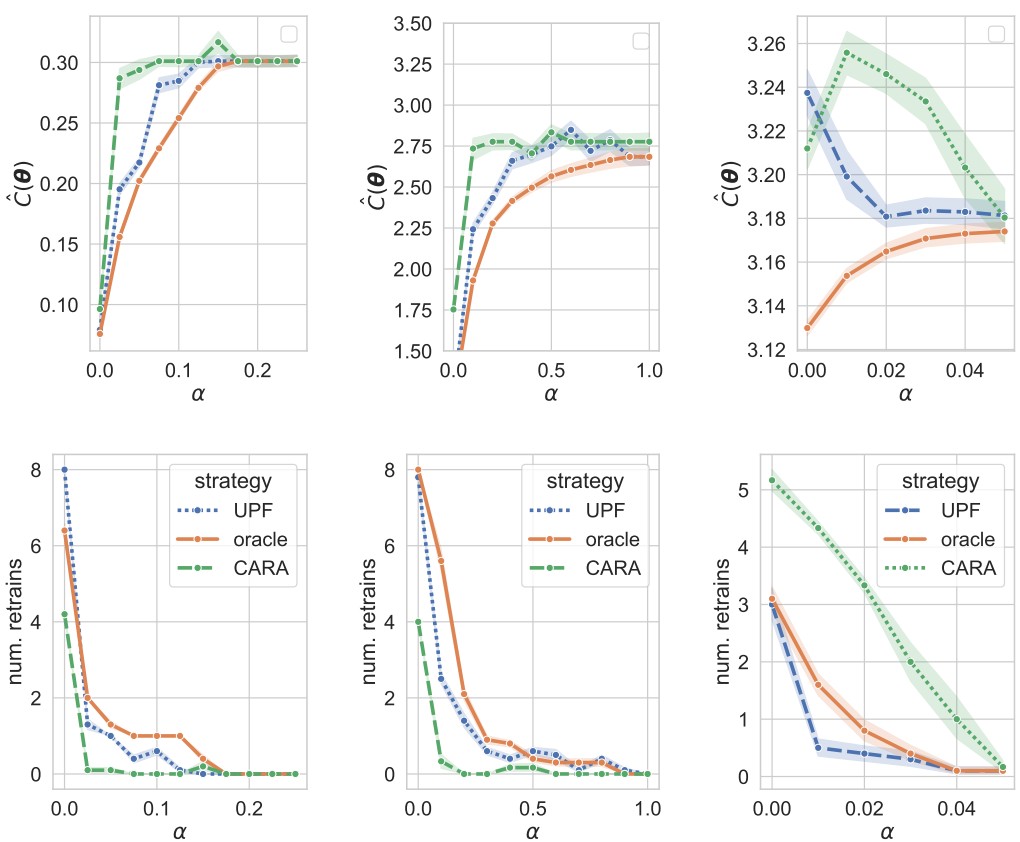

Figure 19: Result on the circles (left), electricity (middle) and epicgames (right) datasets. **Top)** Cost $\hat{C}_\alpha(\boldsymbol{\theta})$ vs $\alpha$. **Bottom)** Number of retrains vs $\alpha$.

### 8.9 METHODOLOGY AS OFFLINE RL

We can frame the retraining problem as an offline RL task (Levine et al., 2020). We define a state space where each state is described by the index of the trained model and the timestep; $S \in \{T\} \times \{T\}$. The action space is to either retrain or not, so $A = \{0, 1\}$. The state transitions are deterministic and known:

$$T(S_{t+1}|S_t = (i,t), A) = \begin{cases} 1 & \text{if } A = 0, S_{t+1} = (i, t+1) \\ 1 & \text{if } A = 1, S_{t+1} = (t+1, t+1) \\ 0 & \text{o.w.} \end{cases} \tag{62}$$

Figure 20 provides a visualization of the MDP. Since the state transitions are deterministic, we can define the deterministic transition function:

$$s_{t+1} = t(a_t, s_t). \tag{63}$$

The reward function only depends on the end state (which describes the performance of a model $i$ evaluated at timestep $t$) and on the action. Using $pe_S$ to denote the performance at a state $S$ and reusing of tradeoff parameter $\alpha$, we have the reward function:

$$r(a_t, s_{t+1}) = -\alpha a_t - pe_{s_{t+1}}. \tag{64}$$

To match our setting, the discount factor has to be set to one $\gamma = 1$.

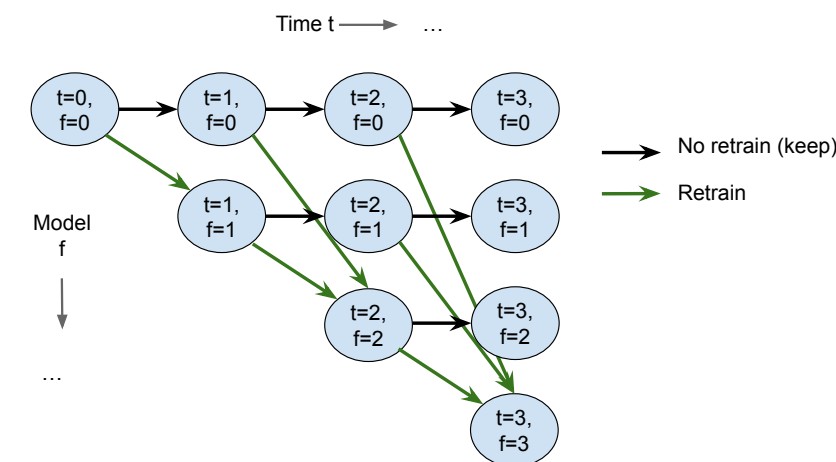

Figure 20: Visualization of the MDP

The goal is to learn a policy $\pi$ on offline data to generalize to the online period. The offline dataset is given by: $\mathcal{D}^{offline} = \{s_n, a_n, r_n\}_{n=1}^N$.

The objective is defined as:

$$J(\pi) = \mathbb{E}_{\tau \sim p_\pi(\tau)}\Big[\sum_{t=w}^{T+w} r(s_t, a_t)\Big], \tag{65}$$

which is the same objective as we defined, with the added option of defining a random policy to make decisions $p_\pi(\boldsymbol{\theta})$:

$$J(\pi) = \mathbb{E}_{\boldsymbol{\theta} \sim p_\pi(\boldsymbol{\theta})}\Big[\sum_{t=w}^{T+w} r(s_t, a_t)\Big] \tag{66}$$

$$= -\mathbb{E}_{\boldsymbol{\theta} \sim p_\pi(\boldsymbol{\theta})}\Big[\sum_{t=w}^{T+w} \alpha a_t + p e_{s_{t+1}}\Big] \tag{67}$$

$$= \mathbb{E}_{\boldsymbol{\theta} \sim p_\pi(\boldsymbol{\theta})}\Big[C_\alpha(\boldsymbol{\theta})\Big]. \tag{68}$$

$Q$-**learning (approximate dynamic methods)** The basic idea of $Q$-learning is to define a $Q$ function and to derive a deterministic policy $\pi$ from it. The $Q$ function is defined as follows;

$$Q^\pi(s_t, a_t) = \mathbb{E}_{\tau \sim p_{\tau|s_t, a_t}}\Big[\sum_{t'=t}^{T+w} r(s_{t'}, a_{t'})\Big] \tag{69}$$

and the policy is set to:

$$\pi(a_t|s_t) = \delta(a_t = \arg\max Q(s_t, a_t)). \tag{70}$$

Since the optimal policy $\pi^*$ should satisfy

$$Q^*(s_t, a_t) = r(s_t, a_t) + \mathbb{E}_{s_t \sim T(s_{t+1}|s_t, a_t)}\Big[\max_{a_{t+1}} Q^*(s_{t+1}, a_{t+1})\Big], \tag{71}$$

one algorithm is to train $Q_\phi$ until that equation is satisfied.

In our case, the transition is deterministic, so we can define $s_{t+1} = t(s_t, a_t)$ and have

$$Q^*(s_t, a_t) = r(s_t, a_t) + max_{a_{t+1}} Q^*(t(s_t, a_t), a_{t+1}). \tag{72}$$

The idea is then to parameterize $Q_\phi$, and minimize the following for all samples in the dataset using the Bellman update:

$$\sum_n (Q_\phi(s_n, a_n) - [r(s_n, a_n) + \max_{a'} Q_\phi(s', a')])^2. \tag{73}$$

First we set the target:

$$y_n = r(s_n, a_n) + \max_{a'} Q_\phi(s', a') \tag{74}$$

then we optimize

$$\frac{\partial}{\partial \phi} \sum_n (Q_\phi(s_n, a_n) - y_n)^2. \tag{75}$$

and the algorithm iterates between those two steps. We can therefore apply any $Q$-learning method to our problem, provided that it uses a standard $Q_\phi$ parameterization.

**Connecting Q-learning to our UPF algorithm**

In our setting, we have special knowledge of the structure of $Q$. First, there is no randomness on the transition state, so we know that:

$$y_n = r(s_n, a_n) + \max_{a_{n+1}} Q_\phi(t(s_n, a_n), a_{n+1}) \tag{76}$$

By definition, we have that:

$$Q_\phi(s_t, a_t) = -a_t \alpha - pe_{s,t} + \max_{a_{t+1}} Q_\phi(t(s_t, a_t), a_{t+1}) \tag{77}$$

While computing the Bellman update and setting the target, we can see that the $Q$ function of one of the last states $Q_\phi(s_{T,x}, \cdot)$ will have to predict the end performance:

$$Q_\phi(s_{T,x}, \cdot) = -pe_{s_{T,x}}, \tag{78}$$
$$= -f_\phi(s_{T,x}). \tag{79}$$

By the DAG structure of the transition function, and since the $\alpha$ value is known, we can parameterize recursively all the $Q_\phi$ functions with shareable components:

$$Q_\phi(s_{T-1,x}, a_{T-1,x}) = -\alpha a_{T-1,x} - f_\phi(s_{T-1,x}) + \max(-\alpha - f_\phi(s_{T,T}), -f_\phi(s_{T,x})), \tag{80}$$

where each $f_\phi(s_{T-1,x})$ is modeling the performance $pe_{s_{T,x}}$ at that given state.

The MSE objective that is traditionally applied (Eqn. 75) can then be decomposed into 2 terms, where one of the terms corresponds to our objective:

$$\mathcal{L} = \sum_n (Q_\phi(s_n, a_n) - y_n)^2 \tag{81}$$

$$= \Big( -\alpha a_{n,x} - f_\phi(s_n) + \max(-\alpha - f_\phi(s_{T,T}), -f_\phi(s_{T,x})) \tag{82}$$

$$- \Big( a_n \alpha + pe_{s_n} + \max_{a_{n+1}} Q_\phi(t(s_n, a_n), a_{n+1}) \Big) \Big)^2 \tag{83}$$

$$= \Big( f_\phi(s_n) - pe_{s_n} + \max(-\alpha - f_\phi(s_{T,T}), -f_\phi(s_{T,x})) + \max_{a_{n+1}} Q_\phi(t(s_n, a_n), a_{n+1}) \Big)^2 \tag{84}$$

$$\mathcal{L} = \sum_n \Big( f_\phi(s_n) - pe_{s_n} \Big)^2 + C. \tag{85}$$

The term $\Big( f_\phi(s_n) - pe_{s_n} \Big)^2$ in the loss function aligns with our objective, as $A_{i,j}$ represents our model's approximation of the performance metric $pe_{i,j}$. Therefore, with this specific parameterization, we can establish a connection between $Q$-learning and our learning method.

However, as noted in the main text, applying existing ORL methods to this problem would not be effective. The problem involves a deterministic transition matrix and a highly structured reward, both of which are uncommon in typical RL settings. Additionally, most RL methods prioritize scalability to large state or action spaces, use complex models, and assume access to plentiful data, making them ill-suited for our scenario. A key requirement for our approach is training efficiency, given our limited performance data and the need for online adaptation as more information becomes available. If the computational cost of deciding when to retrain is comparable to the retraining process itself, the approach becomes impractical.

### 8.9.1 Offline RL baselines

In this section, we present results using an offline RL baseline that is appropriate for low-data settings: Least-Squares Policy Iteration (LSPI) (Lagoudakis & Parr, 2003). We follow the detailed RL formulation as previously presented. To implement LSPI, we use the model index $i$ and timesteps $t$ as states (following the formulation from the previous section). In LSPI, various approximation methods are introduced to solve the linear equation, but these are unnecessary in our case, as we can solve it exactly due to the small size of our problem. We present various versions of this baseline by changing the $\lambda$ parameter. In Table 7, we can see that this proposed baseline is not competitive. These initial results for this basic formulation of the offline RL problem indicate that more care and design should be taken to appropriately solve this problem using offline RL, supporting that existing RL methods, as they are, may not be well-suited to solve the problem.

Table 7: AUC of the combined performance/retraining cost metric $\hat{C}_\alpha(\boldsymbol{\theta})$, computed over a range of $\alpha$ values, for all datasets. The bolded entries represent the best, and the underlined entries indicate the second best. The $*$ The $*$ denotes statistically significant difference with respect to the next best baseline, evaluated using a Wilcoxon test at the $5\%$ significance level.

|  | electricity | Gauss | circles | airplanes | yelpCHI | epicgames | iWild |
|---|---|---|---|---|---|---|---|
| ADWIN-5% | 2.8099 | 0.4533 | 0.0753 | 2.6353 | 0.1298 | 0.3217 | 3.7371 |
| ADWIN-50% | 2.8131 | 0.4848 | 0.0753 | 2.7147 | 0.1298 | 0.3238 | 4.2564 |
| KSWIN-5% | 3.8979 | 0.3975 | 0.0753 | 3.2300 | 0.1322 | 0.3420 | 4.4268 |
| KSWIN-50% | 4.0521 | 0.9530 | 0.0794 | 3.2042 | 0.1655 | 0.3537 | 4.4268 |
| FHDDM-5% | 3.1525 | 0.3893 | 0.0753 | 2.6577 | 0.1324 | 0.3298 | 4.4267 |
| FHDDM-50% | 3.4037 | 0.5918 | 0.0772 | 2.7077 | 0.1450 | 0.3389 | 4.4268 |
| CARA cumul. | _2.7147_ | 0.3862 | 0.0731 | 2.2900 | 0.1299 | 0.3228 | 3.8922 |
| CARA per. | 2.8986 | 0.4678 | 0.0800 | 2.4061 | 0.1318 | 0.3260 | _3.7527_ |
| CARA | 2.7198 | _0.3841_ | _0.0726_ | **2.2753*** | _0.1294_ | _0.3202_ | 3.9506 |
| LSPI $\lambda = 1$ | 4.3820 | 1.0530 | 0.2412 | 3.7140 | 0.1493 | 0.3523 | - |
| LSPI $\lambda = 0.5$ | 4.5260 | 1.0837 | 0.2455 | 3.6924 | 0.1442 | 0.3566 | - |
| LSPI $\lambda = 0.0$ | 4.5317 | 1.0933 | 0.2478 | 3.5862 | 0.1378 | 0.3573 | - |
| UPF (ours) | **2.5782*** | **0.3829*** | **0.0668*** | _2.2865_ | **0.1293*** | **0.3189*** | **3.0498*** |
| oracle | 2.4217 | 0.3724 | 0.0627 | 2.2298 | 0.1275 | 0.3170 | 2.4973 |

### 8.10 Relating our objective to the CARA formulation

In (Mahadevan & Mathioudakis, 2024), even though they are also tackling the retraining problem, they are formulating the problem differently.

Instead of using a binary vector to model the retraining decisions, they use a sequence of model indices $S = [s_1, \dots, s_T]$ with the constraint that $s_t \in \{0, \dots, t\}$. If $s_t = t$, it signifies a retrain.

The cost objective they consider is similar to ours; they sum over the timesteps to get the cumulative total cost. The cost per timestep is encoded in an upper triangular matrix $C$:

$$C[t', t] = \begin{cases} \bar{\Psi}_{t,t'} \text{ if } t' < t \\ \kappa \text{ if } t' = t \text{ (cost of retraining)} \\ \infty \text{ o.w.} \end{cases} \tag{86}$$

where $\bar{\Psi}_{t,t'}$ is defined as some "relative staleness cost". The total cost is defined as:

$$C^{cara}(S) = \sum_{t=1}^{T} C[s_t, t]. \tag{87}$$

The staleness cost is defined as the cost of using a model $f_1$ to classify data from $Q_2$, approximated by dataset $D_3$:

$$\Psi(Q_2, D_3, f_1) \triangleq \sum_{q \sim Q_2} \frac{1}{|D_3|} \sum_{x,y \sim D_3} sim(q, x) \ell(f_1, x, y) \tag{88}$$

The aim of this metric is to predict the performance of $f_1$ on the query points in $Q_2$ by computing the loss on a reference dataset $D_3$. The idea is to weight the loss at each sample of $D_3$ by how similar they are to the query samples in $Q_2$ (this is the role of $sim(q,x)$).

$$\ell\big(f_3(q), y_q\big) \approx \frac{1}{|D_3|} \sum_{x,y \sim D_3} sim(q,x)\ell(f_1,x,y) \tag{89}$$

$$\Psi(Q_2, D_3, f_1) \approx Ne\mathbb{E}_{Q_2}[\ell(f_3(X),Y)] \tag{90}$$

$$\approx Nepe_{t3,t2} \tag{91}$$

The relative staleness cost is defined as the difference between staleness costs:

$$\bar{\Psi}_{t,t'} = \Psi(Q_t, D_t, f_{t'}) - \Psi(Q_t, D_{t'}, f_{t'}). \tag{92}$$

This is intended to approximate the relative gap of performance:

$$\bar{\Psi}_{t,t'} \approx Ne(pe_{t',t} - pe_{t,t}) \tag{93}$$

In our experiment, we directly use $\Psi(Q_t, D_t, f_{t'})$ as an approximation of $pe_{t',t}$ and apply the CARA algorithm directly on the staleness costs instead of using the relative staleness cost.

**Relating it to our formulation**   Our objective is given by;

$$C(\theta) = c\|\theta\|_1 + eN\sum_{t=1}^{T} pe_{r_\theta,t}. \tag{94}$$

To understand the connection with our formulation, we start by rewriting the CARA cost as:

$$C^{cara}(S) = \sum_{t=1}^{T} \mathbb{1}[s_t = t]\kappa + \mathbb{1}[s_t < t]\bar{\Psi}_{t,s_t} \tag{95}$$

$$= \sum_{t=1}^{T} \mathbb{1}[s_t = t]\kappa + \mathbb{1}[s_t < t]\bar{\Psi}_{t,s_t} \tag{96}$$

$$\approx \sum_{t=1}^{T} \mathbb{1}[s_t = t]\kappa + Ne\mathbb{1}[s_t < t]\big(pe_{s_t,t} - pe_{t,t}\big) \text{ from equation 93} \tag{97}$$

$$C^{cara}(\boldsymbol{\theta}) = \kappa\|\boldsymbol{\theta}\|_1 + Ne\sum_{t=1}^{T}\big(pe_{r_{\boldsymbol{\theta}},t} - pe_{t,t}\big) \text{ switching to our notation with } \boldsymbol{\theta}. \tag{98}$$

This reveals the assumptions that are required for both solutions to coincide. First, this approximation for the loss of a future model $f_t$ should hold:

$$\ell\big(f_t(x_q), y_q\big) \approx \frac{1}{|D_t|} \sum_{x,y \sim D_t} sim(x_q,x)\ell(f_1,x,y) \tag{99}$$

Second, in order to have:

$$C(\theta) = C^{cara}(\boldsymbol{\theta}) \tag{100}$$

we need

$$\kappa = c + \frac{Ne\sum_{t=1}^{T} pe_{t,t}}{\|\boldsymbol{\theta}\|_1}. \tag{101}$$

**Proof:** We require that:

$$c\|\boldsymbol{\theta}\|_1 + Ne\sum_{t=1}^{T} pe_{r_{\boldsymbol{\theta}},t} = \kappa\|\boldsymbol{\theta}\|_1 + Ne\sum_{t=1}^{T}\big(pe_{r_{\boldsymbol{\theta}},t} - pe_{t,t}\big). \tag{102}$$

This implies that:

$$c\|\boldsymbol{\theta}\|_1 + Ne\sum_{t=1}^{T} pe_{r_{\boldsymbol{\theta}},t} = \kappa\|\boldsymbol{\theta}\|_1 + Ne\sum_{t=1}^{T} pe_{r_{\boldsymbol{\theta}},t} - Ne\sum_{t=1}^{T} pe_{t,t}, \tag{103}$$

and hence that:

$$\kappa = c + \frac{Ne\sum_{t=1}^{T} pe_{t,t}}{\|\boldsymbol{\theta}\|_1}. \tag{104}$$

The cost of retraining $\kappa$ in the CARA formulation must thus scale with the minimum performance cost that can be obtained by always using the most recent model $Ne\sum_{t=1}^{T} pe_{t,t}$, divided by the number of retrains that have been made. It is of course not possible to set $\kappa$ to this value, as it depends on $\boldsymbol{\theta}$, but it gives insight into how the formulations relate to each other.

## 8.11 VARYING TRAINING DATA SIZE

In this section, we provide experimental results where we assume that we have access to fewer offline time steps and analyze how it impacts the results. We display the relative improvement of the best baseline vs. the competing baselines by reporting normalized AUC values in Tables 8,9, and10. Overall, our method remains effective in scenarios with reduced training data. It demonstrates greater robustness compared to the CARA baselines, which can be explained by the fact that it can adapt to new information received during the online process, which CARA cannot do. With very few training steps ($w = 2$), the CARA baselines suffer the most, reaching more than twice the error for some datasets. With more data ($w = 4$), the relative performance is more in line with larger datasets ($w = 7$), with UPF remaining the best.

Table 8: $w = 2$. Normalized AUC of the combined performance/retraining cost metric $\hat{C}_\alpha(\boldsymbol{\theta})$, computed over a range of $\alpha$ values, for all datasets. We normalize by dividing by the best value for each dataset. The bolded entries represent the best. The $*$ denotes statistical significance with respect to the next best baseline, evaluated using a Wilcoxon test at the $5\%$ significance level.

| $w = 2$ | electricity | airplanes | yelpCHI | epicgames | Gauss | circles |
|---|---|---|---|---|---|---|
| CARA | **1.0000** | 1.0101 | 1.0100 | 1.0282 | 2.6519 | 1.4792 |
| CARA c. | 1.0669 | 1.0680 | 0.0544 | 2.7437 | 4.0150 | 1.6872 |
| CARA per. | 2.1971 | 1.6703 | 0.0661 | 2.9131 | 10.6965 | 1.8901 |
| UPF | 1.0258 | **1.0000*** | **1.0000*** | **1.0000** | **1.0000*** | **1.0000*** |

Table 9: $w = 4$. Normalized AUC of the combined performance/retraining cost metric $\hat{C}_\alpha(\boldsymbol{\theta})$, computed over a range of $\alpha$ values, for all datasets. We normalize by dividing by the best value for each dataset. The bolded entries represent the best. The $*$ denotes statistical significance with respect to the next best baseline, evaluated using a Wilcoxon test at the $5\%$ significance level.

| $w = 4$ | electricity | airplanes | yelpCHI | epicgames | Gauss | circles |
|---|---|---|---|---|---|---|
| CARA | 1.0093 | 1.0024 | **1.0000** | 1.0063 | 1.0049 | 1.0653 |
| CARA per. | 1.1029 | 1.0721 | 1.0017 | 1.0168 | 1.0984 | 1.0045 |
| CARA c. | 1.0153 | 1.0060 | 1.0025 | 1.0220 | 1.0042 | 1.0501 |
| UPF | **1.0000*** | **1.0000*** | 1.0008 | **1.0000*** | **1.0000*** | **1.0000*** |

Table 10: $w = 7$. Normalized AUC of the combined performance/retraining cost metric $\hat{C}_\alpha(\boldsymbol{\theta})$, computed over a range of $\alpha$ values, for all datasets. We normalize by dividing by the best value for each dataset. The bolded entries represent the best. The $*$ denotes statistical significance with respect to the next best baseline, evaluated using a Wilcoxon test at the $5\%$ significance level.

| $w = 7$ | electricity | airplanes | yelpCHI | epicgames | Gauss | circles |
|---|---|---|---|---|---|---|
| CARA c. | 1.0530 | 1.0065 | 1.0046 | 1.0122 | 1.0086 | 1.0944 |
| CARA per. | 1.1244 | 1.0575 | 1.0193 | 1.0223 | 1.2219 | 1.1976 |
| CARA | 1.0549 | **1.0000*** | 1.0008 | 1.0041 | 1.0031 | 1.0868 |
| UPF (ours) | **1.0000*** | 1.0050 | **1.0000*** | **1.0000*** | **1.0000*** | **1.0000*** |

## 8.12 PRELIMINARY RESULTS ON THE WILD TEMPORAL DATASET

In this section, we present preliminary results on one dataset from the suite of temporal datasets from Yao et al. (2022). Specifically, we present preliminray results from the **yearbook** dataset.

To construct our sequence of datasets $\mathcal{D}_t, \ldots,$ we follow the construction from (Yao et al., 2022). For training, we iteratively add more samples from each year, spanning from 1930 to 2012. For testing, we evaluate only on samples from the most recent year. As for the model $f_t$, we use the ERM model from (Yao et al., 2022), and follow the training procedure from Yao et al. (2022). We use a similar setup to the one followed in our experiment, setting the offline window size $w = 7$, evaluating over an online phase of $T = 8$ steps, and presenting results over 10 trials (See table 11). Preliminary results for this dataset which can be seen in Table 12 are inline with the results from the main paper.

Table 11: Dataset description. $w$ denotes the number of timestep of the offline phase, $T$ denotes the number of timestep of the online phase. The Model describes the architecture used for each $f_t$.

| Dataset | Model | $\alpha_{max}$ | w | $|\mathcal{M}_{<0}|$ | T | Dataset size ($|D|$) | Num. features | Task |
|---|---|---|---|---|---|---|---|---|
| **yearbook** | ERM | 0.5 | 7 | 21 | 8 | (varies) | 32X32X3 | Binary |

Table 12: AUC of the combined performance/retraining cost metric $\hat{C}_\alpha(\boldsymbol{\theta})$, computed over a range of $\alpha$ values, for all datasets. The bolded entries represent the best, and the underlined entries indicate the second best. The $*$ The $*$ denotes statistically significant difference with respect to the next best baseline, evaluated using a Wilcoxon test at the $5\%$ significance level.

|  | yearbook |
|---|---|
| CARA cumul | 0.0351 |
| CARA per. | 0.0195 |
| CARA | 0.0322 |
| UPF | **0.0120**$*$ |
| Oracle | 0.0105 |

## 8.13 LIST OF TIMM PRETRAINED VISION MODELS

```
'beit_base_patch16_224',
'beitv2_base_patch16_224',
'caformer_s18',
'cait_s24_224',
'cait_xxs24_224',
'cait_xxs36_224',
'coat_lite_mini',
'coat_lite_small',
'coat_lite_tiny',
'coat_mini',
'coat_tiny',
'coatnet_0_rw_224',
'coatnet_bn_0_rw_224',
'coatnet_nano_rw_224',
'coatnet_rmlp_1_rw_224',
'coatnet_rmlp_nano_rw_224',
'coatnext_nano_rw_224',
'convformer_s18',
'convit_base',
'convit_small',
'convit_tiny',
'convmixer_1024_20_ks9_p14',
'convnext_atto',
'convnext_atto_ols',
```

```
1836        'convnext_base',
1837        'convnext_femto',
1838        'convnext_femto_ols',
1839        'convnext_nano',
1840        'convnext_nano_ols',
1841        'convnext_pico',
1842        'convnext_pico_ols',
1843        'convnext_small',
1844        'convnext_tiny',
1845        'convnext_tiny_hnf',
1846        'convnextv2_atto',
1847        'convnextv2_femto',
1848        'convnextv2_nano',
1849        'convnextv2_pico',
1850        'convnextv2_tiny',
1851        'crossvit_15_240',
1852        'crossvit_15_dagger_240',
1853        'crossvit_15_dagger_408',
1854        'crossvit_18_240',
1855        'crossvit_18_dagger_240',
1856        'crossvit_9_240',
1857        'crossvit_9_dagger_240',
1858        'crossvit_base_240',
1859        'crossvit_small_240',
1860        'crossvit_tiny_240',
1861        'cs3darknet_focus_l',
1862        'cs3darknet_focus_m',
1863        'cs3darknet_l',
1864        'cs3darknet_m',
1865        'cs3darknet_x',
1866        'cs3edgenet_x',
1867        'cs3se_edgenet_x',
1868        'cs3sedarknet_l',
1869        'cs3sedarknet_x',
1870        'cspdarknet53',
1871        'cspresnet50',
1872        'cspresnext50',
1873        'darknet53',
1874        'darknetaa53',
1875        'davit_base',
1876        'davit_small',
1877        'davit_tiny',
1878        'deit3_base_patch16_224',
1879        'deit3_medium_patch16_224',
1880        'deit3_small_patch16_224',
1881        'deit_base_distilled_patch16_224',
1882        'deit_base_patch16_224',
1883        'deit_small_distilled_patch16_224',
1884        'deit_small_patch16_224',
1885        'deit_tiny_distilled_patch16_224',
1886        'deit_tiny_patch16_224',
1887        'densenet121',
1888        'densenet161',
1889        'densenet169',
            'densenet201',
            'densenetblur121d',
            'dla102',
            'dla102x',
            'dla102x2',
```

```
'dla169',
'dla34',
'dla46_c',
'dla46x_c',
'dla60',
'dla60_res2net',
'dla60_res2next',
'dla60x',
'dla60x_c',
'dm_nfnet_f0',
'dm_nfnet_f1',
'dpn68',
'dpn68b',
'dpn92',
'dpn98',
'eca_nfnet_l0',
'eca_nfnet_l1',
'eca_nfnet_l2',
'eca_resnet33ts',
'eca_resnext26ts'
```

