# OpenReview forum: "When to retrain a machine learning model"
_ICLR.cc/2025/Conference — Submitted to ICLR 2025_

### Official Review · Reviewer_XDtg · 2024-11-02

**Soundness:** 2
**Presentation:** 2
**Contribution:** 3
**Rating:** 6
**Confidence:** 3

**Summary:**

This paper focuses on retraining problem i.e., when to retrain or update the machine learning model during its development. The authors provide a theoretical formulation of the retraining problem. Then, they propose a novel retraining decision strategy based on performance forcasting. Besides, the authors also give detailed discussion about the relationship between the proposed formulation and offline reinforcement learning. Experiments on tabular and image datasets are conducted to support the proposed method.

**Strengths:**

1. The investigated problem is newly proposed and important in practice. Determining when to retrain or update the machine learning model is of great importance in many real-world applications.
2. The authors provides a rigorous formulation of the retraining problem. The analysis about upper limits on the optimal number of retrain based on performance bounds is novel and sound in my view.

**Weaknesses:**

First of all, I would like to acknowledge that I am unfarmilar with the related works. From my reading of the paper, I have the following concerns:
### Major
- Some essential related works are not well compared in this work. For example, in section 4.1, the authors propose to learn a performance predictor to forecast unknown entries in PE. As far as I can tell, the motivation of such predictor is very close to uncertainty estimation/quantification [1] [2]. However, the paper do not discuss the difference bettween the predictor in 4.1 with this line of works.
- Strict assumptions. In line 217, the authors assume that all the dataset is IID and there is no distribution shift. Such assumption seems inpratical to me. If there is no distribution shift, why the performance of machine learning model drops as time goes by? I suspect the IID assumption is relatively too strong for the retrain problem this paper focuses on. Besides, I also have concerns if the learned performance predictor described in 4.1 can work well under distribution shifts. Since it is not easy to aquire such a predictor in OOD detection or error detection tasks according to [1] [2].
- Limited experiments. The authors conduct experiments on both tabular and image datasets under binary classification settings. The number of samples, features and classes are all very small in these datasets. Thus I have concerns about whether the proposed method is practical on more realistic benchmarks.
### Minor
- The retrain problem may have been mitigated to some extent by techniques from other fields. For example, life-long learning and transfer learning. If this is true, the importance of this paper will be weaken. Some discussion about this point will benefit the overall quality of this paper.

It is possible that I missed some details of this paper. I am pleased to be corrected by the authors.

[1] A Baseline for Detecting Misclassified and Out-of-Distribution Examples in Neural Networks, ICLR'17

[2] Energy-based Out-of-distribution Detection, NIPS'20

**Questions:**

See above.

---

> ### Author Response · Authors · 2024-11-20
> **Rebuttal XDtg**
>
> Thank you for the review of the paper. We will update a revised version of the paper before the end of the rebuttal with the changes and suggestions.
>
> ## Missing essential related work (uncertainty estimation/quantification)
>
> Uncertainty quantification methods estimate the probability of error for individual samples given a model $f$ with the goal to assess the risk of specific predictions ($P(f(x)=y|x)$). In contrast, our approach forecasts the model's overall future performance (e.g. accuracy) of known and unknown models to determine optimal retraining points, with the goal to minimize total costs by balancing inference errors and retraining cost. We do not consider the uncertainty of model predictions in our framework. Therefore, the cited work could not be used as baselines, and are not closely related to our problem.  We agree that making that difference clear in the text is important, so as suggested we will clarify the difference with this line of work and add in the text:
>     `` This forecasting problem can seem similar to the problem of uncertainty quantification [1][2], but we are targeting average performance of unknown models, not the probability of error of a given model at a given input $P(f(x)=y|x)$.
>
> ## Strict assumptions
>
> ( We assume the reviewer meant line 247 Eq.8 ). This is a misunderstanding and we will add the word ``conditionally'' to clarify, i.e., the random variables $A_{i,j}$ are conditionally independent of one another given the distributional parameters $\alpha$ and $\beta$. We do not make that assumption and the reviewer is correct to state that it would be an unrealistic assumption to make.
> Line 247 assumes that the target variables $A_{i,j}$ \textbf{conditioned} on the parameters are independent of one another. This is not the same as stating that the $A_{i,j}$ are IID. The equations states that the dependencies between the variables $A_{i,j}$ and $A_{k,l}$ are captured by the dependencies between the parameters $\alpha(x_{i,j}),\beta(x_{i,j})$ and $\alpha(x_{k,l}),\beta(x_{k,l})$.   Specifically, we \textbf{do not} make the assumption that $P(A_{i,j}, A_{k,l}) = P(A_{i,j})P(A_{k,l})$.
>
> ## Limited experiments.
> We have made a typo in the table detailing our experiment. The iWild dataset is actually a 100-class classification task with 224x224 resolution images and 1 categorical feature, and is of reasonable size. The purpose of adding this dataset was to ensure that we did indeed present a realistic large-scale experiment, as we recognized, alongside the reviewer, that such experiments are important to assess the practicality of the work. We have corrected this typo in the Appendix, and we hope that this addresses concerns about the practicality of the method. We are also working towards adding an additional dataset from the same family as the iWild dataset which will add before the end of the rebuttal period if time permits.
>
> ## (Minor) Relation to life-long learning and transfer learning
> We group life-long learning and transfer learning with the fields of adaptive learning and online learning that we mentioned in our text. These represent similar alternative approaches to the general problem of using machine learning models in production and addressing the inevitable distribution shift. We agree that, in principle, they could constitute a sensible solution, as the reviewer suggested. The main reason we view these methods as not directly applicable is that they generally focus only on optimizing performance without considering the full cost of modifying a model. This approach is sensible in some settings where updates are not too costly; for example, companies do update their recommender systems every day with incremental learning methods. However, in other settings (safety-critical or with regulatory issues), the cost of changing or modifying a model in production can go beyond the number of gradient updates or sample complexity. For example, costs associated with deployment and risk assessment of a fraud detection model will be incurred no matter how small the update to the model is.
>
> If we were to address this by integrating these baselines and applying a flat cost each time one of the models is modified, it would be unfair to them, as it would be completely orthogonal to the intended use of these algorithms.
>
> To clarify, we will modify the text in line 62 to include those additional related fields:
> ``Other related areas are online or adaptive learning (Hoi et al., 2021) and life-long learning which
> updates models with a continuous stream of data through gradual gradient updates, and transfer learning which adapts model from one distribution to another. However, this
> differs from our problem, as it focuses on maximizing performance while abstracting the practical
> retraining costs involved in production deployment. In practice, the cost of retraining can go beyond
> the number of gradient updates or sample complexity, as discussed above.''

---

> > ### Comment · Reviewer_XDtg · 2024-11-25
> >
> > I thank the authors for their response. There are some additional questions that I would like to confirm with the authors.
> >
> > - our approach forecasts the model's overall future performance (e.g. accuracy) of known and unknown models to determine optimal retraining points
> >
> > What is the relationship between ``forecasts the model's overall future performance'' in this paper and [1] [2]. I suspect that these two papers are closely related and they are also closely related to uncertainty estimation in essence.
> >
> > [1] Energy-based Automated Model Evaluation, ICLR'24
> >
> > [2] The Entropy Enigma: Success and Failure of Entropy Minimization, ICML'24

---

> > > ### Author Response · Authors · 2024-11-28
> > >
> > > Thank you for reading our response.
> > > We have also added another larger-scale experiment in Appendix 8.12 to address your earlier point regarding the limited number of experiments.
> > >
> > > To answer your question, the relationship between our approach and the performance estimation (PE) methods[1][2] is that both aim to predict a model's future performance. However, the core difference lies in the information available to perform this task.
> > >
> > > PE methods assume access to the trained model $f_i$ and partial access to the future distribution $D_t$ through features $X_t$, enabling them to design prediction functions like $pe(i,t) = g(f_i, X_t, \text{other info})$. In contrast, our setting lacks access to both $f_i$ and $D_t$. Instead, we only have past performance data and training/evaluation timestamps, so our function must be structured as $pe(i,t) = g(i,t, \text{other info})$.
> > >
> > > This lack of access to $f_i$ and $D_t$ requires a fundamentally different approach. While both methods aim to forecast model performance, PE methods generally rely on inputs unavailable in our setting, making their direct application infeasible.
> > >
> > > To clarify this distinction, and to address how these methods may initially appear more closely related than they actually are, we have added a discussion on this point in our revised version. This expanded discussion appears in the extended related work section in the Appendix.

---

> > > > ### Comment · Reviewer_XDtg · 2024-11-28
> > > >
> > > > Thanks for your reply. I believe my main concerns have been addressed. As a result, I will increase my confidence from 2 to 3 accordingly.
> > > >
> > > > To my best knowledge, this work is focusing on a quite novel research problem. It is crucial to explain the relationship between the so-called retraining problem and other exists concepts like online adaption (as also mentioned by reviewer Pv7F). In my opinion, a semial work does not need to check every box. It is completely fine to show its value in a limited setting.

---

> > > > > ### Author Response · Authors · 2024-11-29
> > > > >
> > > > > We thank the reviewer for their response and for raising the confidence score! To ensure we address any remaining concerns, could the reviewer clarify which weaknesses or areas for improvement they believe still persist?

---

### Official Review · Reviewer_Pv7F · 2024-11-03

**Soundness:** 3
**Presentation:** 2
**Contribution:** 3
**Rating:** 5
**Confidence:** 5

**Summary:**

Authors consider the problem how deciding when to retrain machine learning model based on the relative importance of accuracy and retraining effort.  The work only considers classification tasks. The proposed solution is based on uncertainty of the future states, and it continuously forecast the model performance to make decisions on whether to retrain or not. The trade-off between accuracy loss and retraining cost is modeled through a hyperparameter \alpha, which allows us to consider different scenarios, e.g., when we are considering LLM (huge retraining cost) or very simple ML models (low retraining cost).

The challenges that the solution aims at solving are (a) how to work with very limited information, (b) how to measure the impact of the distribution drift, (c) how to specify the trade-off between retraining and poor performance.

**Strengths:**

- The work focuses on a timely problem that has not been studied enough in the literature. The problem of deciding when to retrain a model is crucial in real deployments of ML models.

- The authors show analytically the relation of their proposed formulation and problem with similar approaches, such as reinforcement learning (cf. 8.6) and CARA formulation (8.7).

- Results in one of the datasets show that the proposed solution (UPF) is able to keep track of the optimal solution when the relation between accuracy and retraining cost varies. This behavior is shown for both total cost and number of retraining instances.

**Weaknesses:**

- Very important: It is not mentioned in the title or main parts of the work (Abstract) that the work focuses on classification only. It is important, as there is no discussion or result on how this solution could work for regression tasks. Please make sure you explicitly mention this, including a possible modification of title.

- It is not clear how the solution solves some of the 3 challenges mentioned in the abstract. A reader would think that the challenges highlighted in the abstract would be solved or at least considered in the work. However, it is not clear how, for example, the solution delas with "1) very limited information". While some results are provided in the appendix, it would be appreciated if the evaluation of how this challenge is handled by the solution is given more importance and mentioned in the main text. For example, how the performance of the proposed solution and the baselines evolves as the amount of the available information decreases would be really helpful.

- Among the limitations mentioned for the most important baseline (CARA), it is said that CARA assumes that "the difficulty of the task remains constant". But it seems that the proposed solution also considers such aspect. The authors fix the complexity of retraining for any time, the number of samples, etc, so the difficulty is not modified throughout the experiment. We can see this in the text between (2) and (3), where the temporal dependance of the retraining cost c, cost loss e, and number of predictions N is dropped.

- Authors mention in the abstract and other times that "online learning formulation overlooks key practical considerations. A proper comparison (in evaluation results) against online learning approaches is missing (as well as against RL, mentioned below). In such way, it would be clear why online learning cannot be used. Is it because it doesn't consider retraining cost? How can we compare retraining cost (of the whole model) with the incremental updates applied in online learning? Why is it discarded? What are the challenges of including it in the evaluation?

- IMPORTANT: Problems with mathematical notation: Some terms are not defined, some terms are called in different ways, and some mathematical assumptions are not well supported. ... The detailed cases are provided in the following box "questions".

- Proposition 3.1 is interesting. However, it is difficult to fully grasp its significance because the result is not shown in the Results Section. Prop. 3.1 provides a bound but we do not know if this bound is tight or so loose that it is meaningless. Some results on the values provided by the bound and the practical values obtained by the algorithms would be extremely appreciated.

- Lack of complexity comparison. The complexity of the algorithm and the corresponding comparison with the baselines is missing. For example, while CARA is evaluated and compared against the proposed algorithm, it is several times mentioned that CARA is computationally intensive (cf. 8.1). Yet, we do not know if the proposed algorithm is similarly demanding or it offers significant improvement in terms of complexity reduction. A quantitative analysis of the complexity of the different solutions would be welcome to strengthen the contribution of the paper.

- While the comparison with Reinforcement Learning (RL) is discussed in the main document and the appendix, highlighting that RL is not sample efficient and is not impractical, and authors show the analytical correspondence of the proposed formulation and the RL formulation, it would have been welcomed if the authors have provided quantitative proofs of such claims. Evaluating a SotA RL algorithm adapted to the problem considered and measuring its performance with the same datasets and use cases would have provided support to the claim of not considering RL for the problem.

- From appendix 8.5, we can see that the selected dataset to show the main results in Section 6 (e.g., fig. 2) is the best case, where the proposed algorithm clearly performs better than the baselines. This behavior is not so evident for the other datasets in 8.5. For example, In Fig. 5, for the yelp dataset, we can see that the proposed algorithm is not among the best performing solutions, and it is overpassed by several cases, some as simple as adwin.

- The results section have some flaws,
    - Why Figure 3 only shows the values of alpha between 0 and 0.1, when previous figures show the range between 0 and 1? Please provide an explanation or extend the range evaluated to match that of the other cases.

- In appendix 8.5, the oracle is not provided. Thus, we miss information about how far are algorithms from the best.

**Questions:**

- UPF, the name of the proposed solution, is not defined

- Mathematical issues:
    - In (1), x_i,j and y_i,j are not defined. While one can infer that they corresponds to input parameters and classification labels, it is not mentioned anywhere. The same happens with X, Y. Moreover, if X, Y represent the set of values in D_j (please explain it), the notation in (2) is not rigorous and should be modified, as the values should be x_i,j and y_i,j that below to D_j. Besides this, x_i,j is later used to refer to a different term. Please do not use the same notation for two different things,  correct this problem.
    - It is not clear why the authors define the matrix PE. The values in PE are never used as a matrix, they are always considered as individual values. Thus, why creating a matrix? one can just define the values as scalars.
    - How does g(t, I_<t) depend on t apart from I_<t? if there is no more dependence, please remove "t,", keep only the second value.
    - There are several inconsistencies in the use of the notation, for example with the sub-index \phi of \alpha.
    -  The whole section 4 is a bit unclear and could benefit from a revision.  The following points are examples of this aspect.
        - g_\phi is not defined but directly mentioned in Section 4.  What is defined before is g() with no sub-index. Thus, it is not clear at first what is the meaning of \phi.
        - PE_i,j in line 240 has a different notation and it is not anymore a matrix. Moreover, two lines below the comma between i and j is dropped, the same happens to A_i,j and _ij. Please be consistent with notation.
        - In (8), shouldnt the left part also respect the i<j  condition on the right part?
        - In (9), please provide a proof of mutual independency.
        - x_i,j is used here for something different than in (1)
         a_ii,j is not defined. What is it? A realization of A_i,j? please be explicit.
        - How accurate is the approximation of Beta as a normal distribution? Could the authors include results for the case in which no approximation is done?
        - Which is the reason for which you can assume that the variance for all x_i,j is constant? (line 267)
        - Again, it is not explained what \phi refers to.
        - Footnote 2 must be in the main text. It is an important aspect, a definition of a key value, and should be in the main text.
        - It is not clear what "l_1 distance between the average feature vectors of the two recent datasets" refers to. x_i,j = [i,j,i-j,z_shift] are referred to as input features. Are the "feature vectors" referring to x_i,j? or do they refer to the input data of the dataset? Please explain better this aspect, maybe re-naming some of these terms.
        - In (13), the notation of P(A_i,j) is not clear. It seems that it refers to a probability of an event A_i,j, but it actually means that the distribution of A_i,j is a Beta distribution. Please explain better this relation, maybe re-denoting P() for other notation.
        - In (14), we do not know the values for times bigger than t, it is not clear how you perform the steps related with forecasting.Shouldnt it be that T in the sum becomes "t"?
        - In line 300, it is misleading that "we only incur the performance cost of the old model", because we also incur in the future cost for the decisions we will make.
        - "W" in (19) is not defined.

- Section 5: Do datasets D_j include all previous datasets? not specified.

- Section 5: C(\theta) sometimes has sub-index alpha, sometimes not. Be consistent.

- Section 5: What is "w".

- Section 5: Line 400, "For \mu...", it is not clear why this is included in dataset. A better explanation of how you use \mu would be appreciated.

- In table 2, bold values are not clearly explained

- Please, revise the "* denotes statistical significance ..." sentences, mostly in tables. some have typos (e.g. Table XX), and some are not clear. If * represents the same for all tables, just use the exact same sentence in each of the tables, to make it clear for the reader. For example, in Table 5, "∗ denotes statistical significance with respect to the next best baseline," probably wants to express that "∗ denotes statistically significant difference with respect to the next best baseline,", as in Table 3, where there is actually a typo ("significanct"). Please be consistent referring to the same concept exactly in the same terms each time it appears.

- Conclusions could be extended to provide some more insights about the contribution, limitations/to-do points, and key take-away messages of the work

------ Typos and writing suggestions ------
- "uncertainly", line 092
- Line 314, "15" -> (15), same for line 325 with 16- 18
- In appendix 8.2, please correct the repeated typo "t1" instead of "t_1"
- In appendix 8.2, please correct the repeated typo "p" instead of "pe"
- In appendix 8.2, please explain what the red color means. Please also explain each of the steps in the derivation.
- "each datasets", appendix 8.4.
- In 8.4, please explain each column in Table 4, especially w, T.
- In 8.5,  please correct the first line "Figures 4,  5 8"
- no comma in "We observe in Table 3,"
- Adwin and Fhddm -> wrong style in line 459

---

> ### Author Response · Authors · 2024-11-20
> **Rebuttal Pv7F 1/4**
>
> We thank the reviewer for their thorough review and comments. The suggested changes and comments are excellent and significantly improved the presentation of our work. We will update a revised version of the paper before the end of the rebuttal with the changes and suggestions.
>
> ## Focus on classification
>
>  While it is true that our experiments showcase only classification tasks, the methodology is not actually tied to the specific case of classification. The only constraint on the type of task that our proposed method can handle as-is is that its performance is *measured by a bounded metric* as we stated in our text on line 274;
>     ``As stated, this parameterization is appropriate for bounded losses''. This is not exclusive to classification. For example, normalized metrics such as the mean absolute percentage error (MAPE) or the normalized mean absolute error (nMAE), which are suitable for regression, could be used as is.
>
> We also agree that our empirical evaluation focuses on classification and should be made clear in the presentation, so we will modify the abstract to reflect this limitation of our work:
>
> ''To address this, we present a principled formulation of the retraining problem and propose an uncertainty-based method that makes decisions by continually forecasting the evolution of model performance *evaluated with a bounded metric*. Our experiments *addressing classification tasks* show that the method consistently outperforms existing baselines on 7 datasets. We thoroughly assess its robustness to mis-specified cost trade-off.''
>
> Moreover, extending our method to non-bounded losses, such as RMSE, could be done by swapping our Beta parameterization for a truncated Gaussian or log-Gaussian distribution.
>
> We allude to this on line 274:
> ``As stated, this parameterization is appropriate for bounded losses. Other distributions can be used to
> model different loss domains if needed''.
>
> However, we agree that this was vague. We will provide a concrete example of how to extend our method to a non-bounded regression metric, such as RMSE, in the appendix to clarify this point.
>
> ## It is not clear how, for example, the solution deals with "1) very limited information"
>
> In the context of machine learning, we consider that a dataset consisting of less than 100 datapoints, (around 30 in our experiment) would be classified under ''very limited information''. The provided results in the appendix is even more limited, as it uses around 3 points for the offline training phase.
> The fact that our solution deals with ``very limited information'' is reflected in our method design in three ways: 1) We incorporate maximum structural knowledge of the problem into our methodology; 2) we use lightweight inference modules, such as logistic regression; and  3) we integrate uncertainty into our model and opt for a risk-averse policy to balance the limited amount of information.
> We appreciate that this is an important point to illustrate, so we will add an extension of this section of the appendix by covering a wider range of cardinality of available information and modify the text to highlight that point.
>
> In particular we will make the following changes;
>     line 85:
>    ''We propose a novel retraining decision procedure based on performance forecasting. Our proposed algorithm is robust and outperforms existing baselines. It requires minimal performance data by fully leveraging the problem structure, employing compact regression models, and balancing the uncertainty caused by data scarcity through an uncertainty-informed decision process.''
>     as well as in
>  line 268: ``Maximizing the likelihood w.r.t. to the
> mean parameters then becomes a standard mean square error objective. Given the expectation of operating in a very low-data regime, we rely on simple inference models, such as linear regression.''
>
> ## Assumption that ''difficulty of the task remains constant''
>
>  We were referring to the fact that under distribution shift ($p_t \neq p_{t+1}$), it is possible that the difficulty of the task itself could fluctuate. This means that even with a constant number of samples $N$, constant cost of retraining $c$ and cost of error $e$, the performance of each newly trained model $f_t$, i.e.,   $E_{D_t \sim p_t} [\ell( f_t(X) ,Y)] $
> could change. The CARA formulation implicitly assumes that this stays constant, meaning that if you were to always retrain, you would always maintain the same performance in expectation. We will clarify that point by modifying the text at line 168;
>     ``We do not assume that the difficulty of the task remains constant, i.e., we do not assume that $PE_{t,t} = PE_{t+1,t+1}$''.

---

> ### Author Response · Authors · 2024-11-20
> **Rebuttal 2/4**
>
> ## Missing comparison with online learning
>
> The key practical considerations missing from the online learning formulation make the problems that online learning methods tackle fundamentally different. Online learning (and related fields) aims to minimize updates to a model to optimize performance, while we focus on assessing whether an update is worth its cost.
>
> Therefore, we cannot make meaningful comparisons to these methods. The reviewer’s question, ``How can we compare retraining cost (of the whole model) with incremental updates in online learning?'' reflects the problem. In our view, updating any model (even with small updates) incurs a flat cost (e.g., deployment, risk assessment, human resources) that should be considered.
>
> We could present an experiment applying a flat cost to each update in an online learning method, but this would be misleading, as these methods were not designed for this scenario.
>
> Instead, we could integrate this mechanism into our framework, where new models $f_t$ can be obtained without retraining from scratch. This is why we note in line 161: ``The datasets and trained models can be formed and obtained through any means depending on the task at hand; for example, $f_1$ could be fine-tuned from $f_0$ and $\mathcal{D}_1$ could contain $\mathcal{D}_0$.''
>
> We could extend our framework to account for varying retraining costs by considering different models ($f^{(1)}_t, f^{(2)}_t, \dots$) with associated costs based on the number of updates applied. For example, $f^{(1)}_t$ could be a full retrain, while $f^{(2)}_t$ could represent a cheaper model fine-tuned from $f_{t-1}$. This is an avenue worth exploring, but would constitute a substantial research endeavour that is beyond the scope of our current work.
>
> Finally, offline RL can be more easily integrated into our framework, so we have provided results on sensible baselines as suggested by the reviewer.
>
> ## Proposition 3.1 results
>
> This is a great suggestion. Of course the tightness of the bound is dependent on how well we can bound the performance gap $L$, but we can provide results with some rough estimates of this quantity to provide a sense of how tight this result is and how it is meant to be used. We will provide such results in the appendix of the revised version. The bound is meant to be used as a first test to determine if someone should even consider retraining their model. If the problem is indeed worth solving, as is the case in the experiments we presented, the bound will not be particularly tight.
>
> ## Lack of complexity comparison
>
> We agree that providing explicit training complexity figures of the baselines would strengthen the presentation. We will add this in the revised version.
>
> ## Comparison with offline RL baselines
>
>  We agree that providing a state-of-the-art offline RL baseline would indeed strengthen this point. However, we are faced with some technical issues when implementing them. For example, COMBO (Yu et al., 2021), which could be viewed as a state-of-the-art offline RL baseline, has more than six hyperparameters to set, and the architecture and training procedure are far too complex for a dataset of only 30 sample points:
>      ``We train an ensemble of 7 such dynamics models ... pick the best 5 models based on
> the validation ... set that contains 1000 transitions in the offline dataset D.
> ... we randomly pick one dynamics model from the best 5 models ... 4-layer feedforward neural network with 200 hidden units...''(Yu et al., 2021).
> In order to implement that baseline, we would have to make numerous ad-hoc adjustments to fit the smaller dataset, which may ultimately lead to a solution that diverges significantly from the intended approach. If time permits, we will add such a baseline within the rebuttal period.
>
> As an alternative, to provide quantitative proof of offline RL methods as requested, we will present a more dated offline RL method (Michail et al., 2003) that is better suited to our setting, as it consists of linear models applied to basis functions. While it still suffers from the listed drawbacks of offline RL, its methodology does not explicitly rely on large validation sets.
>
> Yu, A. Kumar, R. Rafailov, A. Rajeswaran, S. Levine, and C. Finn, “Combo: Conservative offline model-based policy
> optimization,” in Neural Information Processing Systems, 2021.
>
>  Michail G. Lagoudakis and Ronald E. Parr. Least-squares policy iteration. J. Mach. Learn. Res., 4:
> 1107–1149, 2003
>
> ## Figure 2 as best case scenario
>
> Figure 2 is not the best case; it is fairly consistent with the exception of the low-cost point at $\alpha=0$ on a single dataset mentioned by the reviewer (we believe the reviewer is referring to the EpicGames dataset in Figure 6). We will add the oracle to these plots as requested, which will show that this is not a cherry-picked result but rather a consistent result across datasets and $\alpha$ values. We will also expand the appendix section to discuss this outlying result.

---

> > ### Comment · Reviewer_Pv7F · 2024-11-22
> >
> > Thank you a lot for your detailed response and the mentioned changes that you are applying to the paper. I will carefully check the new additions and clarifications when they are available.
> >
> > I have two brief comments.
> >
> > - Suggestion on "## Missing comparison with online learning" part of the response: this is also related to the answer to Reviewer  Eyq7 regarding "Focus on retraining vs adaptation": Apart from moving the footnote to the main text. Make it more explicit. It would improve the readability of the work if you directly add the sentence of your response: *There is no constraint on how the ''retrained'' model ft is obtained. It can be obtained through fine-tuning from a previous model ft−1, adapted, trained from scratch, or any other procedure.* Making such statement clear  and not too late in the paper strenghten the contribution of the work and prevents readers from thinking the solution does not work if you dont fully retrain.
> >
> > You did not respond to the following point (in *italics*). I think it is necessary to improve this explanation to better grasp what's the meaning of z_shift.
> > - *It is not clear what "l_1 distance between the average feature vectors of the two recent datasets" refers to. x_i,j = [i,j,i-j,z_shift] are referred to as input features. Are the "feature vectors" referring to x_i,j? or do they refer to the input data of the dataset? Please explain better this aspect, maybe re-naming some of these terms.*

---

> > > ### Author Response · Authors · 2024-11-23
> > > **Response Reviewer Pv7F**
> > >
> > > Thank you for your response. We have uploaded a revision of the paper and integrated your suggested change. Regarding the second point, we have included details on $z_{shift}$ in Appendix 8.5.

---

> > > > ### Author Response · Authors · 2024-12-02
> > > >
> > > > As the rebuttal period is coming to an end, we ask if the reviewer could indicate whether our response satisfactorily addresses the concerns raised. We believe that we have carefully and thoroughly addressed each point raised in the review and appreciate any additional feedback.

---

> ### Author Response · Authors · 2024-11-20
> **Rebuttal Pv7F 3/4**
>
> ## Flaw in results  (Figure 3)
>
> The range of the tested alpha value is set per dataset, from $\alpha=0 , \dots, \alpha_{max}$. $\alpha_{max}$ is set following the procedure detailed in the experiment section, line 354: ``The upper bound, $\alpha_{max}$, is determined by the $\alpha$ value at which the oracle reaches 0
> retrains''.
> For that specific dataset, that value was $\alpha_{max} = 0.1$.
>
> To clarify this, we will change the label axis of the figure to reach $\alpha_{max}$ instead of $0.1$ and add the explicit $\alpha_{max}$ value per dataset in Table 4.
>
> ## Missing oracle
>
> We initially omitted the oracle because there were too many lines, making the plot difficult to read. We will add it for completeness and increase the size of the plots to improve readability.
>
> ## UPF not defined
>
> This is a mistake, UPF stands for uncertainty performance forecaster. We will add the following when we describe our method in line 234; ``Since this is infeasible, we propose to 1) model these future values as random variables and learn their distributions; and 2) base our decisions on the predicted distributions to construct our method, the Uncertainty-Performance Forecaster (UPF).''
>
> ## Undefined $x_{i,j}$, ..
>
>  Thank you for pointing those out, we will add explicit definition of $X,Y$ and $x_{i,j}, y_{i,j}$ in the revised version. We agree that our notation was confusing, we used the bolded $x_{i,j}$ to differentiate from $x_{i,j}$ but this is confusing, we replaced it for another different notation $r$ altogether.
>
> ## Why PE is presented in matrix notation
>
> That is right, we changed it to a scalar notation as proposed.
>
> ## How does $g(t, I_<t)$ depend on $t$ apart from $I_<t$?
>
> This notation was to highlight the information you have each time you make a decision, i.e. whatever past information you have access to paired with the current time you are. It is true that $t$ can simply be included in $I$.
>
> ## $g_{\phi}$ is not defined but directly mentioned in Section 4.
>
> $\phi$ is introduced as the parameters of our algorithm. To make this explicit, we will modify line 232 from ``A retraining decision algorithm must specify the decision functions $g_{\phi}(t, I<t)$ used to build
> the decision vector $\theta$ '' to ''A retraining decision algorithm must specify the decision functions $g_{\phi}(t, I<t)$ (where $\phi$ contains the parameters of the algorithm) used to build
> the decision vector $\theta$  ''
>
> ## Notation issue with $PE$ and $A$
>
> Thank you for pointing these out, we will correct those typos.
>
> ## Condition in Eq. 8
> Yes, to make this explicit we modified the notation in the revised version.
>
> ## Mutual independency proof (9)
> Eq 9 assumes that the distribution of the target variables $A_{i,j}$ *conditioned* on the parameters is IID.  We will make this assumption explicit in the text by modifying line 247:``Given the parameters $\alpha, \beta$, we model the
> random variables to be independent of each other;''
>
> ## $x_i,j$ is used here for something different than in (1) $a_{ij}$ is not defined. What is it? A realization of $A_{i,j}$? please be explicit.
>
> As state previously, the bolded $x_{i,j}$ are different than the $x_{i,j}$ in 1). We modified it by using an other symbol to make the distinction clearer. $a_{i,j}$ are defined as observation in line 257, we modified this line to; ``From the offline data, we have access to observations of $a_{i,j}\sim A_{i,j}$ ''
>
> ## Beta approximation
>
> The approximation is pretty close. We included results without the Beta approximation in the Appendix.
>
> ## Constant variance assumption
>
> In practice it is probably not constant, this is a modeling choice we made as we have access to a limited amount of training data. We will clarify that this is modeling choice by changing line 267 to : ``As we have access to a limited quantity of data, we make the modeling choice of parameterizing the variance as a constant $\sigma_{\phi}(x_{i,j} ) = \sigma_{\phi}$ ''
>
> ## Footnote 2 must be in the main text.
>
> We will move it in the main text and add additional clarification in the appendix.
>
> ## the notation of $P(A_i,j)$ is not clear
>
> $ P(A_{i,j})$ is the distribution over the bounded performance metric. $Beta$ denotes the pdf of the Beta distribution, we will clarify this in the text.
>
> ## Shouldn't it be that T in the sum becomes "t"?
>
> Eq. 14 describes the random variable that we are aiming to predict, so it includes value that are bigger than $t$.
>
> ##  misleading that "we only incur the performance cost of the old model", because we also incur in the future cost for the decisions we will make.
>
> It was in the context of the decision at a single step $t$. Either we incur the cost the performance of the new model + the cost of retraining, or we incur the cost of performance of the old model.
>
> ## Undefined $W$ (19)
> This is a typo that we have corrected.

---

> ### Author Response · Authors · 2024-11-20
> **Rebuttal Pv7F 4/4**
>
> ## Do datasets $D_j$ include all previous datasets?
>
> In our experiments, no, they do not. For the iWild experiment, we do some overlapping so the dataset $D_j$ contains half of the dataset $D_{j-1}$.  We describe the construction of the datasets in Appendix 8.4 and will clarify that point.
>
> ## $C(\theta)$ sometimes has sub-index alpha, sometimes not.
>
> In some places, we have omitted the $\alpha$ to lighten the notation. We will add it to be more consistent as suggested.
>
> ## What is "w".
>
> $w$ is the number of timesteps in the offline data and is defined in the problem setting line 141.
>
> ## Clarifying $\mu$
>
> How we use $\mu$ is described in the methodology in line 267``Maximizing the likelihood w.r.t. to the
> mean parameters $\mu_{\phi}$ then becomes a standard mean square error objective. Once these parameters are learned, we can recover the corresponding $\alpha,\beta$ parameters
> to obtain our predictive distribution;'' . We will add the explicit connection between the mean and variance parameter and the $\alpha, \beta$ parameters of a Beta distribution in the text.
>
> ## Bolded values in Table 2
>
> Bolded values are the lowest cost, as stated in the table 1 description line 380; ``The bolded entries represent the best, and the underlined entries indicate the
> second best'' . We can repeat this in the caption of Table 2 as well.
>
> ## Please, revise the "$*$ denotes statistical significance ...
>
> We will modify the text following that suggestion.
>
> ## Conclusions could be extended
>
>  the conclusion was indeed cut a little short because of space constraint, we will make space and extend it to integrate additional limitations and future work.
>
> ## Other typos and suggestions
>
> Thank you for pointing those out, we will correct them and integrate the suggestions.

---

### Official Review · Reviewer_sbKs · 2024-11-04

**Soundness:** 2
**Presentation:** 2
**Contribution:** 1
**Rating:** 3
**Confidence:** 4

**Summary:**

The paper addresses the challenge when to retrain ML models in real-world settings with distribution shifts. To this end they introduce an uncertainty-aware approach for performance estimation and continuously forecast model performance.

**Strengths:**

Performance estimation under distribution shift and the the effect of model retraining is an important task and the authors introduce an interesting uncertainty-aware PE approach and draw connections to RL.

**Weaknesses:**

Unfortunately the evaluation of the proposed approach is very limited and the author missed connections to several relevant research streams that have been working on this (or closely related tasks). Therefore closely related baselines are missing and dedicated datasets that were introduced for this task are not used for evaluation.

More specifically, the authors do not discuss the clear connection to the active testing literature and to the whole field of label-free performance estimation. A meaningful presentation of the approach would require an in-depth discussion of the related work from these fields and, importantly, a thorough benchmarking especially to PE methods like [1-3]. Owing to the connection to active risk estimation (and active learning) it would also be interesting to evaluate which data-points exactly need to be labelled for re-training.

As for datasets, the WILD-time benchmark [4] was designed for the very task the authors address and the proposed method should be evaluated on the datsasets curated in this paper.

In addition, I find the evaluation lacks important details, eg how often is re-trained with the different baselines for a given cost (and what is accuracy).




[1] Garg, S., Wu, Y., Balakrishnan, S.,
and Lipton, Z. (2020). A unified view of label shift
estimation. Advances in Neural Information Pro-
cessing Systems, 33:3290–3300.

[2] Guillory, D., Shankar, V.,
Ebrahimi, S., Darrell, T., and Schmidt, L. (2021).
Predicting with confidence on unseen distributions.
In Proceedings of the IEEE/CVF International Con-
ference on Computer Vision, pages 1134–1144.

[3] Chen, M., Goel, K., Sohoni, N. S.,
Poms, F., Fatahalian, K., and R´e, C. (2021). Man-
doline: Model evaluation under distribution shift. In
International conference on machine learning, pages
1617–1629. PMLR.

[4] Yao, H., Choi, C., Cao, B., Lee, Y., Koh, P. W. W., & Finn, C. (2022). Wild-time: A benchmark of in-the-wild distribution shift over time. Advances in Neural Information Processing Systems, 35, 10309-10324.

**Questions:**

See above

---

> ### Author Response · Authors · 2024-11-20
> **Rebuttal sbKs 1/2**
>
> Thank you for the review of the paper. We will update a revised version of the paper before the end of the rebuttal with the changes and suggestions.
>
> ##  Limited evaluation of the proposed approach
>
> We presented results on 7 datasets (2 synthetic and 5 real) with 6 baselines (3 variants from CARA and 3 detection shift baselines). Seven datasets is a reasonable number for assessing performance and is within the norm for strong evaluation. While 6 benchmarking baselines may seem low, this is to be expected as this problem has been largely understudied in the literature (as pointed out by all reviewers). It is therefore to be expected that fewer relevant works are available. The reviewer pointed out three works they believe constitute relevant and important benchmarks that should have been included. We explain below why these are not appropriate baselines.
>     For these reasons, we do not agree that our evaluation and experimental section is limited.
>
> ## Missed connections to several relevant research streams (active testing and label-free performance estimation).
>
> Some aspects of our problem do have connections to the field of active testing and label-free performance estimation, which we will discuss and add to our extended related work section. However, there are fundamental differences in the problem formulation that makes these methods at best tangentially related, and definitely unsuitable to be used as baselines.
>
>
> To restate, our method has two components: a) the performance forecaster module; and b) the retraining decision module.
>
> Active testing, label-free estimation, and the cited works are unsuitable for two reasons:
> 1) These methods are only related to the performance forecasting component of our method. They do not provide a mechanism to produce retraining decisions.
> 2) Even if we were to disregard the first point (which is critical since the goal is to make retraining decisions), and consider the proposed baselines as a replacement for the first component of our method, they still would not be suitable. For most of the performances that we need to forecast, we \textbf{do not} assume access to the underlying datasets, nor the future models.
>
>     To illustrate the second point, we now discuss each of the proposed baselines.
>
> The cited work [1] considers the problem of estimating the shifted label distribution of a target distribution that undergoes label shift from a source distribution, given samples from the source distribution $(X_s, Y_s) \sim P_s$, label-free samples from the target distribution $X_t \sim P_t(X)$, a trained model on $P_s$, $\hat{f}$, and the assumption that the conditional distribution remains constant $(P(Y_s|X_s) = P(Y_t|X_t))$.  Ignoring the fact that this last assumption is already disqualifying for our retraining setting on its own, we also do not have access to most of the models $\hat{f}$ or input data samples $x$.
>     The cited work [3] relies on assumptions similar to those in [1] but aims to predict the performance on the target distribution instead of the target distribution itself.
>
> The other cited work [2] makes fewer assumptions on the target distribution and [2] also produces uncertainty quantification of its prediction. However, [2] also assumes that it has access to input samples and the trained model to perform the prediction.
>
> In summary, we do not see how active testing or label-free performance estimation methods could be used as baselines in our work. Not only do they not include a retraining decision module, but the assumptions that are commonly adopted prevent them from even being used in the forecasting stage. If the reviewer can provide more details concerning how to circumvent these issues, we would be happy to integrate them as additional baselines, as we do believe that more baselines would strengthen the work.
>
> ## Missing dedicated dataset
>
>  We agree that the proposed dataset [4] would be a good dataset for us to include and if time permits, we will include preliminary results before the end of the rebuttal period.
>    However, it is not correct that ``WILD-time benchmark [4] was designed for the very task the authors address''. The WILD-time benchmark dataset was introduced for benchmarking robustness to temporal distribution shift, not for the retraining problem.

---

> > ### Author Response · Authors · 2024-11-20
> > **Rebuttal sbKs 2/2**
> >
> > ## In addition, I find the evaluation lacks important details, eg how often is re-trained with the different baselines for a given cost (and what is accuracy).
> >
> >  We did provide how often the model is re-trained for each baseline, and for each cost that we considered in Appendix 8.5. We also provided the number of retrains and accuracy for specific methods, baselines, and two given values of $\alpha$ in the main text in Table 2 and Figure 2.
> > See line 427 in the main text; ``To gain better insight into the behavior of the different algorithms and how they are impacted by varying retraining cost parameters, we provide a detailed overview for one dataset with two values of $\alpha$: one where the cost of retraining is low and one where it is high, as shown in Table 2. Figure 2 depicts how the the total cost $C^{\theta}$ and the number of retrains vary as $\alpha$ is changed. Appendix 8.5 contains the complete set of results and figures.''

---

> > ### Comment · Reviewer_sbKs · 2024-11-25
> >
> > Thank you. I look forward to seeing results on WILD-time; I was indeed referring to temporal distribution shift, which is the setting your algorithm is designed for.
> >
> > As for baselines, [2] and [3] can easily serve as plugin performance estimator, which is the main contribution of this submission - it would be crucial to report results of these highly-cited baselines, even if you then argue that your method may be more versatile in some respect.

---

> > > ### Author Response · Authors · 2024-11-28
> > >
> > > Thank you for your response.
> > > We have included preliminary results on one dataset from the WILD-Time paper in Appendix 8.12.
> > >
> > > As for the second point, we would first like to emphasize that while the performance estimator is an important part of our work, it is not the main contribution of this submission.
> > > Regarding the baselines you are mentioning,  maybe there is a misunderstanding. In our response, we did not argue that our method was more versatile than [2][3], we explained why [2][3] **could not** be used as plugin performance estimator in our setting.
> > >
> > > The fact that [2] or [3] assumes access to the trained model $f_t$ and that we don't have access to $f_t$ in our setting does not make our method more versatile, it means that [2] and [3] cannot be used to solve our task. Please note, the unavailability of this model is *fundamental* to the task of deciding when to retrain - it is not an assumption or a convenience or a special case.
> > >
> > > In order to decide whether to retrain or not at a time $t$, and obtain $f_t$,  we  don't have access to the model $f_t$ before making our decision. The whole point of the retraining problem is to decide beforehand if it is worth obtaining the retrained model $f_t$ or not, so assuming that we have access to $f_t$ in order to make that decision renders the whole exercise pointless.
> > >
> > > The method in [2], for example, does require access to the model for which it estimates the performance.
> > >
> > > In particular, [2] targets the problem of estimating the accuracy of a **known** model $F$, which was trained on dataset $B$, on some new distribution $T$ (See Section 3 of the paper. The method in [2] also handle cases of disjoint label space, which we will ignore, because we don't consider such cases in our setup.)
> > >
> > > The approach in [2] starts by introducing average confidence $AC$, which is defined as the average over the maximum probability given by model $F$ to each sample of some dataset $B$ in Eqn. 4:
> > > $AC_B = \frac{1}{|B|} \sum_{x \in B} max_{K} F(x) $.
> > >
> > > Notice that the model $F$ for which the performance is being estimated is **required for us to evaluate the proposed metric**.
> > >
> > > The paper then proposes that the difference of average confidences between datasets $B$ and $T$ can be used as a feature to predict future performance in $T$, in Eqn. 6: Difference of Confidences $DoC_{B,T} = AC_B - AC_T$.
> > >
> > > Therefore, in order to use [2] to predict the performance that would be achieved if we were to retrain and thus obtain $f_t$, we would need $f_t$.
> > >
> > > Moreover, our method does not only forecast the performance of the model $f_t$ if it were to be retrained at time $t$. In order to make decisions about whether to retrain, we  forecast the performance of **future** models on **future** datasets as well. These are, by definition, inaccessible. So the method from [2], which requires access to 1) the dataset used to train $f$,  2) $f$, and 3) the dataset used to evaluate $f$, could not be used as a plugin estimator in our method as the reviewer is suggesting.
> > >
> > > The cited paper [3] suffers from the same issue. (See Section 2.2 of that paper, where they define the same problem as in [2]. The problem description from [3] starts with ``We are given a fixed model $f_{\theta} : X \to Y$ , .. '')
> > >
> > > We hope that this clarifies why we are not using these methods as baselines.
> > >
> > > [2] Guillory, D., Shankar, V., Ebrahimi, S., Darrell, T., and Schmidt, L. (2021). Predicting with confidence on unseen distributions. In Proceedings of the IEEE/CVF International Con- ference on Computer Vision, pages 1134–1144.
> > >
> > > [3] Chen, M., Goel, K., Sohoni, N. S., Poms, F., Fatahalian, K., and R´e, C. (2021). Mandoline: Model evaluation under distribution shift. In International conference on machine learning, pages 1617–1629. PMLR.

---

> > > > ### Author Response · Authors · 2024-12-02
> > > >
> > > > As the rebuttal period is coming to an end, we ask if the reviewer could indicate whether our response satisfactorily addresses the concerns raised. We believe that we have carefully and thoroughly addressed each point raised in the review and appreciate any additional feedback.

---

> > > > > ### Comment · Reviewer_sbKs · 2024-12-02
> > > > > **Meaningful baselines are still missing**
> > > > >
> > > > > I appreciate your response. My big concern here is that you focus entirely on comparisons with different flavours of CARA [1] a very recently proposed approach that hasn't received much attention. As I mentioned in my initial review, you have overlooked the entire field of label-free performance estimation - methods such as ATC (Garg et al 2022) are simple to implement, widely used (hundreds of citations) and could easily have been included as baseline. As is, you have conclusively shown that you improve on different flavours of CARA, but given the maturity of the field of label-free performance estimation this is not enough empirical evidence for me.
> > > > > As for importance reweighting and DoC, I don't really follow your argument - the idea would be to use the model f_{t-1} to estimate performance at t and then retrain if required.
> > > > >
> > > > > [1] Ananth Mahadevan and Michael Mathioudakis. Cost-aware retraining for machine learning. Knowledge-Based Systems, 293:111610, 2024

---

### Official Review · Reviewer_Eyq7 · 2024-11-06

**Soundness:** 2
**Presentation:** 3
**Contribution:** 2
**Rating:** 6
**Confidence:** 3

**Summary:**

The paper studies the problem of retraining a machine learning model in practical settings. In specific, this paper considers another dimension when making decisions – the retrain cost. To make decisions in an online setting, the proposed method learns a performance predictor to predict future model performance given some data shift features. Then the retraining decision will be made based on the future performance estimation and the retrain cost. Experiments on five public classification datasets demonstrate the effectiveness of the method.

**Strengths:**

- The paper clearly explains the targeting objective function and the proposed method.
- The studied problem seems new when the new retraining cost dimension is in the constraints.
- The writing is clear and easy to follow.

**Weaknesses:**

There seems to be some unstated assumptions/explanations.
- The proposed method comprises at least two modules, one performance predictor and one decision module. The paper doesn’t discuss or analyze 1) how the predictor’s performance affects the final results, and 2) what the performance requirement of the predictor is.

The related work needs more discussion.
- The paper claims the retraining cost is a new dimension and sets a hyperparameter to control the retraining rate. However, previous methods like change point detection methods often also have a hyperparameter that controls the sensitivity of detecting distribution shifts, which is similar to the effect of the alpha parameter used in the paper. The authors may need to compare their alpha parameter to the sensitivity parameters in change point detection methods, e.g., Adams and MacKay 2007 and Li et al 2021, conceptually or practically.

[Adams and MacKay. Bayesian online changepoint detection. 2007]
[Li et al. Detecting and adapting to irregular distribution shifts in bayesian online learning. NeurIPS 2021]

Besides,
- How does the method demonstrate the claim “have access to only a few examples”? Could you quantify what you mean by "a few examples" and to provide experimental results showing performance with varying amounts of data?
- The paper focus on completely retraining. Could you explain why your method framework refrains from adaptation? Detect-and-adapt sounds reasonable to me as well. Even more, adaptation is more data efficient. So could you discuss the tradeoffs between your approach and adaptation methods, maybe in the aspects of the computational costs or data efficiency.

**Questions:**

Here are some minor comments:
- Shouldn't the cost of the offline training phase also be incorporated? Maybe a discussion on how incorporating offline training might affect the method's performance is helpful.
- Does the performance predictor suffer from distribution shifts and therefore affect the method's performance?
- Please justify the range of the tested alpha values in the robustness study (fig 3), which only have a range of 0.1. A broader range is more helpful to evaluate the robustness of the method.

---

> ### Author Response · Authors · 2024-11-20
> **Rebuttal Eyq7 1/3**
>
> Thank you for the review of the paper. We will update a revised version of the paper before the end of the rebuttal with the changes and suggestions.
>
> ## Unstated assumptions
>
> In relation to the forecasting of the performance module, the underlying assumption is that the future performance of models follows some trend that can be forecasted. In short, the assumption is that the performance of a model trained at time $t$ has some temporal autocorrelation with the performance of a model trained at another time $t+1$, and that the performance of a model evaluated at time $j$ has some temporal autocorrelation with the same model evaluated at another time $j+1$.  this assumption was not clearly outlined, and we agree that adding it would strengthen the presentation.
>
> We will add the following text in Section 4, line 324:  ``To make perfect decisions, we would need future performance values. This is infeasible; however, we assume that there is an underlying temporal autocorrelation between the performance of different models trained at different times, which we aim to exploit to build a predictive model. We therefore propose to 1) ...''
>
> ## Predictor's performance effect on the final result and Performance requirement
>
> This is a great suggestion. We can certainly provide more insights into the performance of the predictor. This will also provide empirical results to comment on the performance requirement of our model as requested.
> Later this week, we will add a dedicated section in the appendix showing the average RMSE of the predicted mean performance in relation to the overall performance of the algorithm.
>
> ## Related work
>
> Thank you for the references; we will add a discussion on the relation of our setting to the change point detection problem.
> In order to transform the changepoint detection problem from Li et al. 2021 to the retraining problem we are considering, we would need to add a cost for adaptation, a cost for accuracy loss, and then introduce an optimization problem to find the best beta value to achieve the optimal number of adaptations. The Li et al. 2021 paper does suggest that the temperature parameter (beta) can be set by the user: ``people can choose different $\beta$ for a specific environment, with a trade-off between adaptation speed and the erased amount of information.''. However, since that beta parameter does not have a specific physical or practical meaning, it is unclear ahead of time exactly how the choice of a particular value is going to lead to different rates of adaptation. Moreover, in our setting, we do not know what this optimal rate of adaptation (retraining frequency) is. Finding this optimal retaining frequency is one of the major difficulties of the retraining problem.
>     Therefore, we would need to develop a mechanism to relate this beta parameter to our alpha parameter which has physical meaning --- it is the cost ratio of retraining over performance. For these reasons, adding the method by Li et al. (2021) as a baseline would require us to design this additional missing component.
> The retraining cost is not a new dimension of the problem; it is an essential component of the problem itself, and without it, a sensible retraining decision algorithm cannot be defined.
>
> ## Could you quantify what you mean by "a few examples"
>
> In practice, the number of models stored along with their recorded performance is likely very limited.  This is  why throughout the work we assume that the number of datapoints available for training is on the order of 10, and refer to this setting as: “have access to only a few examples”. Accordingly, we set the experiment parameters to stay in that low-data regime. By setting the number of offline steps to $w=7$, we collect the performance of $7$ models evaluated on each of $7$ timesteps. Therefore, the total size of the offline data is $|\mathcal{M}_{<0}| = \frac{7 \times 6}{2} = 21$ examples.
>
> To clarify in the text, we add the following in the description of the offline and online data, in Section 3.1 Offline and Online Data, line 179:
>
> ``We assume that we have access to
> all the datasets and trained models during the offline period. In practice, the number of models and datasets is typically limited to only a few (around 10 to 20 at most), which is why we characterize this problem as being in a low-data regime.''
> and will add explicit $|\mathcal{M}_{<0}| $ values in our table detailing the experiments.

---

> ### Author Response · Authors · 2024-11-20
> **Rebuttal Eyq7 2/3**
>
> ## How does the method demonstrate the claim “have access to only a few examples”?
>
> This is reflected in our method design in three ways: 1) We incorporate maximum structural knowledge of the problem into our methodology; 2) we use lightweight inference modules, such as logistic regression; and  3) we integrate uncertainty into our model and opt for a risk-averse policy.
>
>    To make this clearer, we will make the following changes;
>     line 85:
>     ``We propose a novel retraining decision procedure based on performance forecasting. Our proposed algorithm is robust and outperforms existing baselines. It requires minimal performance data by fully leveraging the problem structure, employing compact regression models, and balancing the uncertainty caused by data scarcity through an uncertainty-informed decision process.''
>     as well as in
>  line 268:`Maximizing the likelihood w.r.t. to the
> mean parameters then becomes a standard mean square error objective. Given the expectation of operating in a very low-data regime, we rely on simple inference models, such as linear regression.''
>
> ## Could you provide experimental results showing performance with varying amounts of data?
>
> We present experimental results for varying amounts of data, including an even smaller dataset in \textbf{Appendix 8.8: Low Training Data}. Additionally, we will supplement these results with further experiments to explore the effect of a the size of the offline training dataset.
>
> ## Focus on retraining vs adaptation
>
>  The paper does not focus on completely retraining. There is no constraint on how the ''retrained'' model $f_t$ is obtained. It can be obtained through fine-tuning from a previous model $f_{t-1}$, adapted, trained from scratch, or any other procedure. We have made that comment explicit in footnote 1 on page 3; ``The datasets and trained models can be formed and obtained through any means depending on the task at hand; for example, $f_1$ could be fine-tuned from $f_0$ and $\mathcal{D}_1 $ could contain $\mathcal{D}_0$.''
>     We will move the footnote to the main body of text to clarify that point.
> To showcase this flexibility, we have presented in our iWildCam experiment a setting where instead of retraining from scratch, we fine-tune pretrained models and select from different architectural choices (See lines 367-377).
> Our method can integrate adaption methods as-is.
>
> ## Shouldn't the cost of the offline training phase also be incorporated? Maybe a discussion on how incorporating offline training might affect the method's performance is helpful.
>
> That is a good point and was also requested by another reviewer. We will add a timing table to compare the training complexity of the offline training of the different methods in an updated version before the end of the rebuttal period.
>
> ## Does the performance predictor suffer from distribution shifts and therefore affect the method's performance?
>
> It depends on the nature of the shift and its impact on overall performance. The role of the performance forecaster is to capture and predict the impact of this shift on the performance, so distribution shift is expected and is not, in itself, bad for your proposed approach. If the distribution shift were to be too sudden, unpredictable, and result in sudden and dramatic variation of performance, then yes, the performance forecaster would be negatively impacted. But in this case, so would any retraining method. The hope is that in practice, the distribution shift is gradual and has a somewhat predictable overall impact on performance, which can then be forecasted with reasonable accuracy.
>     To illustrate this point, see Figure 1 of the paper. The fact that the accuracy of a model decreases as time advances (see any line of the same color that tracks the performance of a given model across time) indicates that there is indeed a distribution shift; otherwise, the performance would stay constant. However, in that figure, you can see that this rate of decrease is more or less regular, and is similar across different models (the rate of decrease of model $f_1$ that was trained on $\mathcal{D}_1$ is similar to that of model $f_2$ that was trained on a more recent dataset $\mathcal{D}_2$). In such a case, the performance forecaster would not be negatively impacted by this type of distribution shift.

---

> > ### Author Response · Authors · 2024-11-20
> > **Rebuttal Eyq7 3/3**
> >
> > ## Justification of the range
> >
> > The range of the tested alpha values is set per dataset, from $\alpha=0 , \dots, \alpha_{max}$. $\alpha_{max}$ is set following the procedure detailed in the experiment section, line 354: ``The upper bound, $\alpha_{max}$, is determined by the $\alpha$ value at which the oracle reaches 0 retrains''.   We introduced that procedure for selecting the value of $\alpha$ for the main performance analysis.
> > It is motivated by not having an extended range where there should be zero retrains (which biases performance assessment towards techniques that retrain less often). For that specific dataset, that value was $\alpha_{max} = 0.1$. To clarify this, we will change the label axis of the figure to reach $\alpha_{max}$ instead of $0.1$ and add the explicit $\alpha_{max}$ value per dataset in Table 4.

---

> > > ### Author Response · Authors · 2024-12-02
> > >
> > > As the rebuttal period is coming to an end, we ask if the reviewer could indicate whether our response satisfactorily addresses the concerns raised. We believe that we have carefully and thoroughly addressed each point raised in the review and appreciate any additional feedback.

---

### Author Response · Authors · 2024-11-23
**Global response**

We thank the reviewers for their thoughtful reviews. All reviewers have stated that the problem we are considering and introducing in our work is important and well-defined. The reviewers have recognized that it is highly motivated, as it is crucial for the deployment of real ML models and has been understudied in the literature.
The reviewers also appreciated the explicit and clear connections we have provided to existing branches of ML.

The main weaknesses highlighted by the reviewers can be grouped into two main points.

First, reviewers brought up various closely related fields of ML and asked why methods from these other fields were not included as baselines. It is worth mentioning that we also recognize that the retraining problem sits at the intersection of many fields and might initially seem addressable by existing methods; however, upon closer inspection, none of these fields fully addresses the practical retraining problem we are tackling.

For that reason, we highlighted the fields we consider most relevant in our Related Work and Extended Related Work sections, namely online learning, adaptation, Bayesian optimization with performance estimation, and offline reinforcement learning, which is the closest, as it can actually be framed within our formulation, as shown in Appendix 8.4 of our submission. Specifically, fields such as online learning, life-long learning, active testing, label-free performance estimation, and change-point detection were mentioned. We have clarified in the text how these methods and problem statements relate to the retraining problem we are considering and explained their limitations, especially for those fields more distant from our focus. For methods closer to our problem—such as offline reinforcement learning, which can be reframed to fit our context as discussed in our appendix—we have provided additional results to show why offline RL methods are not well-suited for this setting.

Another point raised was to clarify how the proposed methods handle low-data scenarios and what this entails. To address this, we have made explicit changes in our introduction and methodology to outline how our method handles these settings. We have also extended our analysis to compare baseline performance with varying sample sizes, further strengthening our claim that our method is well-suited for low-data regimes.

We have responded to each point raised by the reviewers with either careful responses and/or concrete changes and additional experiments in the revised version. In particular, we have; 1) extended the related work in Appendix 8.1 to mention the fields raised by the reviewers; 2) added an illustration of how to use Proposition 3.1 in practice in Appendix 8.2.1; 3) added an analysis of how the performance of the performance forecaster impacts the overall algorithm in Appendix 8.5; 4) described how to extend our framework to one-sided bounded metrics, such as RMSE, in Appendix 8.6; 5) shown the impact of the Beta approximation in Appendix 8.6.1; 6) added training complexity in Appendix 8.7; 7) added oracle results in Appendix 8.8; 8) added results for an offline RL baseline in Appendix 8.9.1; and 9) extended our analysis of the impact of varying training sizes in Appendix 8.11, as well as changed the text following the suggestions of the reviewers. The main changes are highlighted in blue.

---

### Meta-Review · Area_Chair_KV7H · 2024-12-17

**Metareview:**

This paper proposes a novel and practical research problem about when to retrain or update the machine learning model. Motivated by this, they present a principled formulation of the retraining problem and propose an uncertainty-aware method to make such a decision. All reviewers agreed that the proposed retraining problem is important, novel, and crucial for real deployments of ML models. Reviewers raised concerns about 1) the relationship with other fields of ML, 2) technical details about the complexity analysis of the offline process, empirical results for proposition 3.1, an offline RL baseline, and potential extension to the regression case. At the end of the rebuttal, there was no consensus, and this came down to a borderline decision. AC oversaw the whole review process. From my reading of this paper, I felt the research problem herein is important, and the proposed method is effective. Seminal papers like this one often struggle in the review process because they don’t fit the conventional fields of methods or applications. However, the total of the reasonable concerns of this manuscript raised by the reviewers are not minor and do require further review before its publication. I encourage the authors to resubmit to other top venues and wish them the best of luck with their work.

**Additional Comments On Reviewer Discussion:**

Reviewers raised many concerns about the relationship with other fields of ML, technical details, and comparison with other baselines. One of the two reviewers with positive views is of low confidence, and the other one did not actively participate in the discussion. Reviewer PVf7 held a negative view and provided a very thorough review. The total of changes required for this manuscript is not minor and does require further review before its publication.

---

### Decision · Program_Chairs · 2025-01-22

Reject